# Substrate-engaged type III secretion system structures reveal gating mechanism for unfolded protein translocation

Sean Miletic[1,2,3,4,5,6], Dirk Fahrenkamp[1,2,3,6], Nikolaus Goessweiner-Mohr[1,3,4,5,6], Jiri Wald [1,2,3,4,5,6], Maurice Pantel[1,2,3], Oliver Vesper[1,2,3,4,5], Vadim Kotov[1,2,3,4,5] & Thomas C. Marlovits [1,2,3,4,5 ✉]

Many bacterial pathogens rely on virulent type III secretion systems (T3SSs) or injectisomes to translocate effector proteins in order to establish infection. The central component of the injectisome is the needle complex which assembles a continuous conduit crossing the bacterial envelope and the host cell membrane to mediate effector protein translocation. However, the molecular principles underlying type III secretion remain elusive. Here, we report a structure of an active *Salmonella enterica* serovar Typhimurium needle complex engaged with the effector protein SptP in two functional states, revealing the complete 800Å-long secretion conduit and unraveling the critical role of the export apparatus (EA) subcomplex in type III secretion. Unfolded substrates enter the EA through a hydrophilic constriction formed by SpaQ proteins, which enables side chain-independent substrate transport. Above, a methionine gasket formed by SpaP proteins functions as a gate that dilates to accommodate substrates while preventing leaky pore formation. Following gate penetration, a moveable SpaR loop first folds up to then support substrate transport. Together, these findings establish the molecular basis for substrate translocation through T3SSs and improve our understanding of bacterial pathogenicity and motility.

[1] University Medical Center Hamburg-Eppendorf (UKE), Institute of Structural and Systems Biology, Hamburg, Germany. [2] Centre for Structural Systems Biology (CSSB), Hamburg, Germany. [3] Deutsches Elektronen-Synchrotron Zentrum (DESY), Hamburg, Germany. [4] Institute of Molecular Biotechnology GmbH (IMBA), Austrian Academy of Sciences, Vienna, Austria. [5] Research Institute of Molecular Pathology (IMP), Vienna, Austria. [6] These authors contributed equally: Sean Miletic, Dirk Fahrenkamp, Nikolaus Goessweiner-Mohr, Jiri Wald. ✉email: marlovits@marlovitslab.org

**M**any important human pathogens including *Salmonella*, *Shigella*, *Yersinia*, and enteropathogenic *Escherichia coli* employ a conserved, virulent type III secretion system (T3SS), also commonly referred to as the injectisome, to deliver a pleiotropic arsenal of proteins into target eukaryotic cells[1]. These proteins modulate host cell signal transduction processes to establish a biological niche within the host, making T3SSs crucial virulence determinants[2]. Yet, the precise mechanisms that allow these secretion systems to facilitate unfolded protein transport across the bacterial envelope and into the host cell while maintaining bacterial membrane remain integrity poorly understood. Therefore, visualizing the translocation process at the molecular level is essential for our understanding of host–pathogen biology and the development of novel therapies targeting bacterial infection.

The injectisome is a large molecular machine, over 3.6 MDa in mass, spanning across the inner and outer bacterial membranes with an extracellular filamentous appendage extending out to target host cells. Chaperones present effector proteins in a non-globular, secretion-competent state to a cytoplasmic sorting platform complex, which sorts and loads effectors into the export apparatus (EA) subcomplex located inside the membrane-bound basal body[3–6]. Extending from the EA is a long, helical needle filament, capped by a tip complex that contacts the host cell membrane via assembly of a translocon pore[7–9]. The basal body and the needle filament, collectively termed the needle complex, function as a continuous conduit for effector protein translocation from the prokaryotic to the host cell cytoplasm[10,11].

Accumulating structural information has revealed a shared common architecture between injectisome and flagellar T3SSs, especially in the EA[6,12–15]. However, in all structures known to date, the proposed translocation channel through the EA is sealed by a gasket with an above loop, making comprehension of substrate transport through the needle complex difficult. Furthermore, it remains unclear how the EA achieves selective effector protein transport given the multitude of proteins that are present in the bacterial cytoplasm.

Visualizing actively secreting injectisomes is however difficult due to the rapid dynamics of protein transport. A pool of protein substrates are translocated through the T3SS in a hierarchical order upon host cell contact, although injectisomes can be artificially induced to secrete proteins in vitro[4,16]. It is unclear what proportion of these injectisomes actively secrete proteins as in *Salmonella*, induced cells can contain tens of needle complexes[10]. Furthermore, translocation is rapid, estimated at a rate of 7–60 molecules per second[17]. With this speed and temporal variability, isolated injectisome needle complexes likely lack protein substrates, or they dissociate during purification procedures. To overcome these hurdles, effector proteins can be artificially trapped in needle complexes by fusion to C-terminal tags resistant to unfolding[18,19]. We previously showed that the *Salmonella* late effector protein SptP fused to a green fluorescent protein (GFP) tag can be visualized as a subtracted density in the needle complex, confirming that the filament functions as the conduit for effector proteins[18]. However, direct visualization of a substrate throughout the complete secretion conduit has remained challenging, leaving questions as to how and where the EA would eventually open to allow passage of effector proteins, while maintaining the integrity and composition of compartments separated by a biological membrane, unresolved.

In this work we report cryo-EM structures of an active *Salmonella enterica* sv. Typhimurium needle complex engaged with a SptP3x-GFP substrate, revealing the complete 800 Å-long secretion conduit and unraveling the critical role of the EA in substrate transport. Unfolded substrates enter the EA through a hydrophilic constriction formed by SpaQ proteins, which enables side chain-independent transport, providing a rationale for the heterogeneity and structural disorder of signal sequences in T3SS effector proteins. Above, a methionine gasket formed by five SpaP proteins functions as a gate that dilates to accommodate substrates but prevents leaky pore formation to maintain the physical boundaries of compartments separated by a biological membrane. Above the gate, a moveable SpaR loop first folds up to then support substrate transport through the needle complex channel. Together, these findings establish the molecular basis for substrate translocation through T3SSs, improving our understanding of bacterial pathogenicity and motility of flagellated bacteria, and pave the way for the development of novel concepts combating bacterial infections.

## Results

**Architecture of the substrate-engaged injectisome.** To obtain molecular snapshots of an injectisome needle complex engaged with a substrate, we applied cryogenic electron microscopy (cryo-EM) to purified needle complexes, which had been enriched for trapped SptP3x-GFP by immunoprecipitation (Supplementary Figs. 1 and 2). Single particle reconstruction provided us with a non-symmetrized density map of the substrate-trapped/engaged needle complex in two active functional states, ranging from 2.4 to 4.5 Å in resolution (Fig. 1, Table 1, Supplementary Figs. 3–5, Electron Microscopy Databank Accession Code: EMD-11781, Protein Data Bank Accession Code: 7ah9 and 7ahi), and resolving a substrate density traversing through the complete secretion path from the cytoplasmic face of the needle complex to the extracellular filament (Fig. 1a, b). The SptP density reveals that the effector protein adopts a non-globular fold during transport through the needle complex (Fig. 1). However, the positional and conformational flexibility of the substrate, propagated throughout the entire translocation path, impeded our efforts to assign specific residues. As a consequence, we modeled the SptP3x-GFP substrate as a polyalanine sequence (Supplementary Fig. 6).

Inside the needle complex, the substrate travels through a secretion conduit built by the EA, a decameric subcomplex made up of three proteins, SpaP (5×), SpaQ (4×), and SpaR (1×), the inner rod composed of $PrgJ_{1-6}$ and the PrgI-containing filament (Figs. 1c and 2a, see Supplementary Table 1 for unified T3SS nomenclature[1]). Together, these proteins form three discrete building blocks that are embedded within three oligomeric protein rings formed by InvG, PrgH and PrgK, a scaffold spanning the two bacterial membranes and the periplasm (Fig. 1a, b and Supplementary Fig. 7).

The EA can be further separated into three discrete sections which form a three-point pseudo-helical interface with the substrate that together consists of two hydrophilic constrictions containing conserved glutamine residues (hereinafter referred to as Q1- and Q2-belt) that sandwich a hydrophobic methionine gasket (hereinafter referred to as M-gate). The substrate enters the EA with its N-terminus through the portal containing the Q1-belt, continues through the M-gate and Q2-belt defining the EA channel, before reaching the atrium chamber of the inner rod and finally the filament tunnel (Fig. 2b).

To be able to investigate the structural changes underlying substrate transport, we also determined the structure of a substrate-free, closed needle complex referred to hereinafter as apo-state (EMD-11780, 7agx). Focused refinement without any symmetry enforcement provided us with a reconstruction yielding an average resolution of ~3.3 Å, resolving the entire EA, the inner rod and parts of the filament (Supplementary Fig. 8). The model that we built is in good agreement with a published apo-state structure of the same complex (6pep) and structurally-related complexes (6r6b, 6r69, 6s3r, 6s3l, 6s3s), together showing (i) closed EAs share a conserved architecture

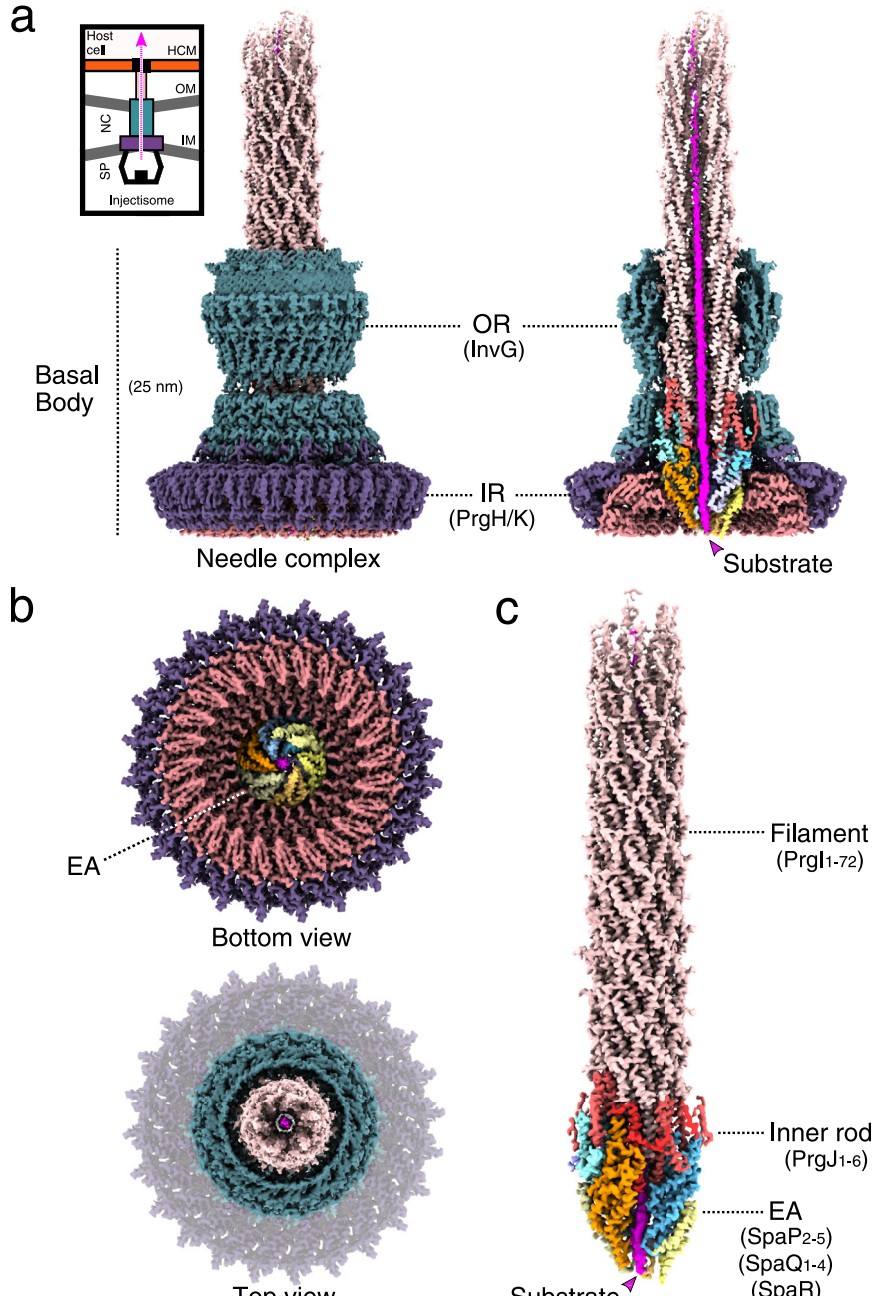

**Fig. 1 Cryo-EM map of the *S. enterica* sv. Typhimurium needle complex engaged with the effector protein substrate SptP3x-GFP. a** The non-symmetrized cryo-EM (C1) reconstruction of the substrate-engaged needle complex (left) with a vertical cross section through the center (right) revealing the substrate shown in magenta throughout the translocation channel. Upper left, is a cartoon schematic of the needle complex injectisome in the bacterial inner and outer membranes (IM/OM) and in contact with a host cell membrane (HCM). NC: needle complex, SP: sorting platform, OR: outer rings, IR: inner rings. The following colors are used in this and all other figures: $PrgH_{1-24}$: dark violet (hex color #6d58ab), $PrgK_{1-24}$: soft red (#dc888d), $InvG_{1-15/16}$: dark cyan (#5f919c), SpaQs: yellow colors ($SpaQ_1$ #FFFF99, $SpaQ_2$ #FFFF66, $SpaQ_3$ #FFCC66, $SpaQ_4$ #CCCC99), SpaPs: blue colors ($SpaP_1$ #99CCFF, $SpaP_2$ #66CCFF, $SpaP_3$ #CCCCFF, $SpaP_4$ #9999FF, $SpaP_5$ #99FFFF), SpaR: orange (#FF9900), PrgJs: red colors ($PrgJ_1$ #FF3333, $PrgJ_{2-6}$ #FF6666), $PrgI_{1-72}$: pale pink (#FFCCCC) and the substrate: magenta (#FF00FF). **b** Top and bottom views of the C1 map showing the export apparatus (EA) and substrate. **c** Cryo-EM map of the filament, inner rod, and the export apparatus components. $SpaP_1$ has been removed to aid visualization of the substrate.

with a defined $5:4:1$ (SpaP : Q : R) stoichiometry and (ii) suggesting that a conserved mechanism orchestrates substrate translocation through these secretion systems (Supplementary Figs. 9 and 10)[6,13,15]. Intriguingly, our apo- and translocation-state EAs superimpose with a low root mean square deviation (rmsd) of 0.49 Å (SpaP : Q : R), revealing that in fact only subtle conformational changes are needed to facilitate substrate

transport through the channel of the needle complex (Supplementary Fig. 11).

**SpaQ_{1-4} form a hydrophilic Q1-belt to engage substrates**. The substrate first engages the needle complex structure through the EA core complex portal formed by four SpaQ proteins, confirming our previous results that the central opening localized at

**Table 1 Cryo-EM data collection, refinement, and validation statistics.**

| | Apo-state (EMDB-11780) (PDB 7agx) | Substrate-trapped state 1 (EMDB-11781) (PDB 7ah9) | Substrate-trapped state 2 (EMDB-11781) (PDB 7ahi) |
|---|---|---|---|
| Data collection and processing | | | |
| Magnification | 130kx | 81kx | |
| Voltage (kV) | 300 | 300 | |
| Electron exposure (e−/Å$^2$) | 31.5 | 53 | |
| Defocus range (μm) | 0.3-5.2 | 0.3-5.2 | |
| Pixel size (Å) | 1.09 | 1.1 | |
| Symmetry imposed | C1 | C1 | |
| Initial particle images (no.) | 234102 | 837325 | |
| Final particle images (no.) | 54491 | 77411 | |
| Average map resolution (Å) | 3.36 | 3.32 | |
| FSC threshold | 0.143 | 0.143 | |
| Map resolution range (Å) | 2.4-4.5 | 2.4-4.5 | |
| Refinement | | | |
| Initial model used (PDB code) | 6s3l, 2lpz | 3gr1, 3gr5, 4g08 | |
| Model resolution (Å) | 3.36 | 3.32 | |
| FSC threshold | 0.143 | 0.143 | |
| Model resolution range (Å) | | | |
| Map sharpening $B$ factor (Å$^2$) | −30 | −30 | |
| Model composition | | | |
| Non-hydrogen atoms | 23669 | 200048 | |
| Protein residues | 3031 | 25300 | |
| Ligands | 0 | 14 | |
| $B$ factors (Å$^2$) | −30 | −30 | −30 |
| Protein | 37.0 | 111.0 | 111.19 |
| Ligand | | 88.0 | 87.5 |
| R.m.s. deviations | | | |
| Bond lengths (Å) | 0.0072 | 0.0094 | 0.0094 |
| Bond angles (°) | 1.16 | 1.21 | 1.21 |
| Validation | | | |
| MolProbity score | 1.05 | 1.02 | 1.01 |
| Clashscore | 2.64 | 2.36 | 2.35 |
| Poor rotamers (%) | 0.08 | 0.00 | 0.00 |
| Ramachandran plot | | | |
| Favored (%) | 98.94 | 98.20 | 98.19 |
| Allowed (%) | 1.06 | 1.80 | 1.81 |
| Disallowed (%) | 0.0 | 0.0 | 0.0 |

the cytoplasmic tip of the EA serves as the substrate entry site[18]. Gln41, Gln43, and Gln45 on four SpaQ loops connecting α-helices α1 and α2, and SpaR Gln208 together form the Q1-belt in the cytoplasmic tip of the EA core complex (Fig. 3a and Supplementary Fig. 12). SpaS, which binds to the EA complex and simultaneously wraps around all four SpaQ loops in the homologous and recombinantly produced EA structure from *Vibrio mimicus* (6s3l), dissociates from the substrate-engaged needle complex during purification, which could explain why SpaQ$_1$, and to a lesser extent SpaQ$_2$, adopt a more open conformation (rmsd: 1.27 Å; Supplementary Fig. 13)[15]. SpaQ homologs of many important human pathogens share a conserved Gln-X-Gln-X-Gln motif within the aforementioned loop, which effectively renders the environment in the Q1-belt hydrophilic (Supplementary Fig. 14). Likewise, conserved hydrophilic residues can be found in the corresponding loop connecting α-helices α1 and α2 in the homologous flagellar FliQ protein, together suggesting that the Q1-belt could play an important role for substrate translocation through virulent and flagellar T3SSs (Supplementary Fig. 15).

Based on the appearance of its density, the substrate is largely unfolded in the Q1-belt area of the translocation path, indicating sequences corresponding to loops and β-sheets in natively folded SptP have mostly been trapped in the EA portal (Figs. 2b and 3b, and Supplementary Fig. 6). Notably, little structural information is available for the very N-termini of injectisome effector proteins,

especially when complexed with their cognate chaperones, supporting prediction models in which these sequences are typically intrinsically disordered[20,21]. Therefore, it is conceivable that our model provides mechanistic insights into how substrate proteins are loaded into the needle complex. In total, the Q1-belt is shaped by 13 glutamines (3× in each of the four SpaQs, 1× in SpaR), which localize within close proximity to the trapped SptP3x-GFP substrate. Out of these, Gln43 of SpaQ$_3$ and SpaQ$_4$ establish hydrogen-bond interactions with the substrate backbone carbonyl oxygens and amine hydrogens (Fig. 3b). Due to the spiral staircase arrangement, the SpaQ/R glutamines provide complementary interaction interfaces over a length of ~20 Å, which, upon successive binding to the substrate, cause an increase in avidity that stabilizes the substrate in the EA portal below the M-gate (Figs. 2 and 3b, c). In agreement with this concept, mutation of the three glutamines to the aliphatic amino acids alanine or glycine, but not serine, impairs secretion of SptP and SipA effector proteins, revealing that the Q1-belt is implicated in substrate translocation (Fig. 3d). In addition, mutation of the Q1-belt could cause assembly defects as seen in the different secretion profiles of the alanine and glycine mutants (Fig. 3d).

Together, our structural data reveals that loading of effector proteins into the needle complex can be accomplished in a side chain-independent manner and hence provides a rationale to explain the structural disorder and plasticity of N-terminal signal sequences observed in injectisome effector proteins.

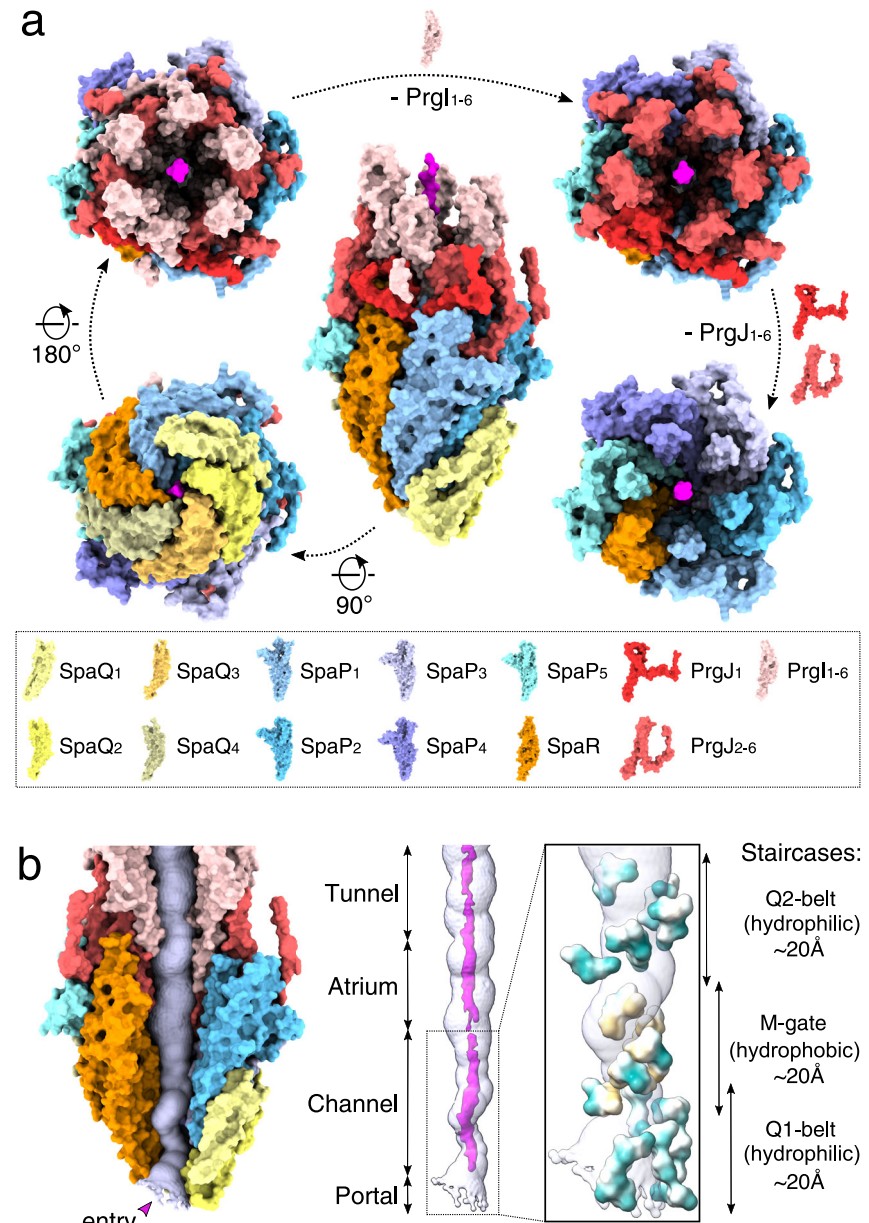

**Fig. 2 The export apparatus (EA) forms a translocation channel for substrates. a** Modular assembly of the substrate-trapped EA, inner rod (PrgJ), and the first six PrgI subunits of the helical filament. Individual protein components are shown in the dashed box below. SptP3x-GFP is shown in magenta. **b** Left: view of the EA with SpaP₁, PrgJ₁-₂, and PrgI₁ removed and the substrate translocation path displayed as a white surface. Right: four discrete sections (portal, channel, atrium, and tunnel) of the translocation path are shown with the EM density corresponding to the substrate (threshold: 0.015). Right box: magnification highlighting surfaces of residues forming hydrophilic and hydrophobic staircases encircling the portal and channel. Green: hydrophilic; white: neutral; gold: hydrophobic.

**SpaP₁₋₅ form a methionine M-gate to regulate, transport and maintain membrane homeostasis.** Following engagement in the hydrophilic Q1-belt, the substrate asymmetrically twists up through the ~5.5 Å-wide M-gate (Fig. 4a, b). The loops between the fifth and sixth α-helix of each of the five SpaP proteins contain three conserved methionines (Met185-187), which, together with conserved SpaR Phe212, form a hydrophobic constriction seen in substrate-free structures[6,12,13,15] (Fig. 4a, b and Supplementary Figs. 12 and 16–19). Similar to the Q1-belt, the pseudo-helical structure of the EA causes these methionines to form a ~20 Å-long, spiral staircase-like gate that shifts to accommodate the substrate passing through (Figs. 2 and 4a, b, and Supplementary Video 1).

Notably, the SpaP pentamer (SpaP₁₋₅) in our substrate-engaged structure superimposes well with its apo-state counterpart (rmsd: 0.45 Å), which reinforces the concept that transport of substrate proteins across the bacterial envelope does not involve large conformational changes but is facilitated by subtle rearrangements that, within the M-gate, are mostly limited to the side chains of the three methionines (Fig. 4a, b and Supplementary Fig. 11 and Supplementary Video 1). Similar to the Q1-belt area, the density corresponding to the trapped substrate fits best to an unfolded polypeptide (Fig. 2b and Supplementary Fig. 6), which, together with the size constraints imposed by the M-gate (~5.5 Å), supports our observation that non-folded SptP sequences have been trapped in the EA.

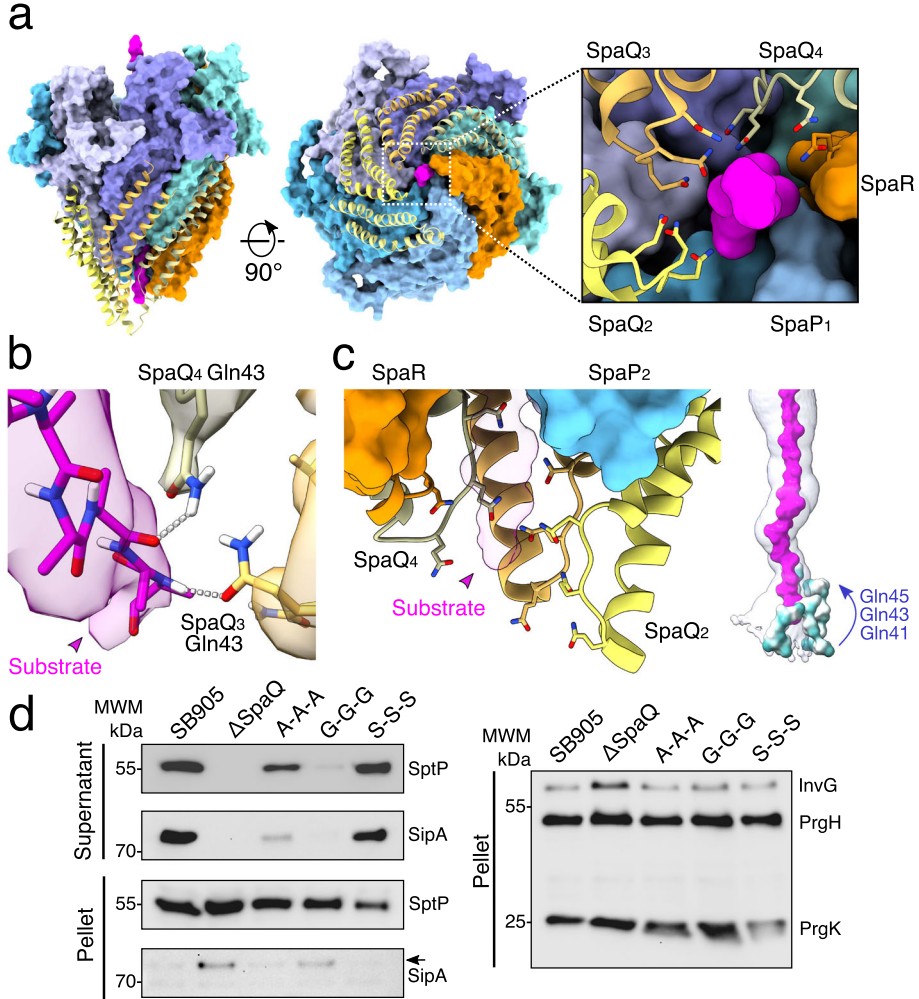

**Fig. 3 The Q1-belt portal of the EA facilitates substrate loading into the needle complex. a** Surface representation of the substrate-engaged EA, with SpaQ$_{1-4}$ shown as yellow ribbon diagrams. Right box: a magnified view of the conserved SpaQ glutamine residues Gln41, Gln43, and Gln45 involved in substrate engagement, depicted in stick representation. **b** Hydrogen-bond formation between the substrate backbone and SpaQ$_3$ Gln43 and SpaQ$_4$ Gln43. The surfaces represent EM density (threshold: 0.014). **c** Side view of the SpaQ Q1-belt displayed as ribbon diagrams encircling the SptP3x-GFP substrate with the side chains of Gln41/43/45 shown in stick representation. Far right: Surface representations of the Q1-belt residues Gln41/43/45 colored by hydrophobicity. Green: hydrophilic; white: neutral; gold: hydrophobic. **d** Representative immunoblots of three independent secretion assays of SptP and SipA effector proteins using SB905 (wild type), SpaQ knockout (ΔSpaQ), and *Salmonella* strains harboring mutations targeting the Q1-belt. Source data for Fig. 3d are provided with this paper.

To functionally characterize the M-gate based on our structural data, we reconstituted a *Salmonella* SpaP-knockout strain (SpaP$^{KO}$) with exogenous SpaP carrying mutations targeting the conserved methionine motif. Negative stain EM revealed that substitution of the motif with glycine and alanine (SpaP$^{GGG}$, SpaP$^{AAA}$), as well as cysteine (SpaP$^{CCC}$), resulted in markedly reduced numbers of needle complexes under injectisome-inducing conditions, demonstrating that needle complex assembly is impaired in these strains (Fig. 4c and Supplementary Fig. 20). As proteins forming the inner rod (PrgJ) and the filament (PrgI) are transported through the virulent T3SS as a natural part of the injectisome assembly process, our data provides evidence that at least the alanine and cysteine mutant strains retain some ability to actively transport proteins through their mutated EAs and hence the methionine motif alone is not strictly required for substrate translocation[22]. However, all three strains display impaired growth kinetics, suggesting that the assembly of EAs, whose M-gates are composed of residues with side chains smaller than methionine, creates pores in the inner bacterial membrane that likely short circuit the membrane

potential and cause reduced pathogen fitness (Fig. 4d and Supplementary Figs. 21 and 22). Consistent with this hypothesis, replacement of the corresponding methionines (Met209-211) in the flagellar homolog FliP to alanines has been shown to increase membrane conductance[23].

To further corroborate this hypothesis, we introduced either tryptophans (SpaP$^{WWW}$) or phenylalanines (SpaP$^{FFF}$) into SpaP to mimic the hydrophobic properties of methionine. Evidently, substitution of the methionine motif with these amino acids did not affect bacterial growth kinetics (Fig. 4d and Supplementary Fig. 21). However, substitution with phenylalanine but not tryptophan restored needle complex assembly to almost wild-type levels, indicating that the bulky hydrophobic side chain of tryptophan effectively prevents leaky pore formation but, compared to methionine and phenylalanine, appears to be too inflexible to efficiently facilitate substrate translocation (Fig. 4c and Supplementary Fig. 20).

Based on our findings, we reasoned that substrates penetrating the methionine network of the EA cause an opening of what appears to be a hydrophobic gate just large enough to

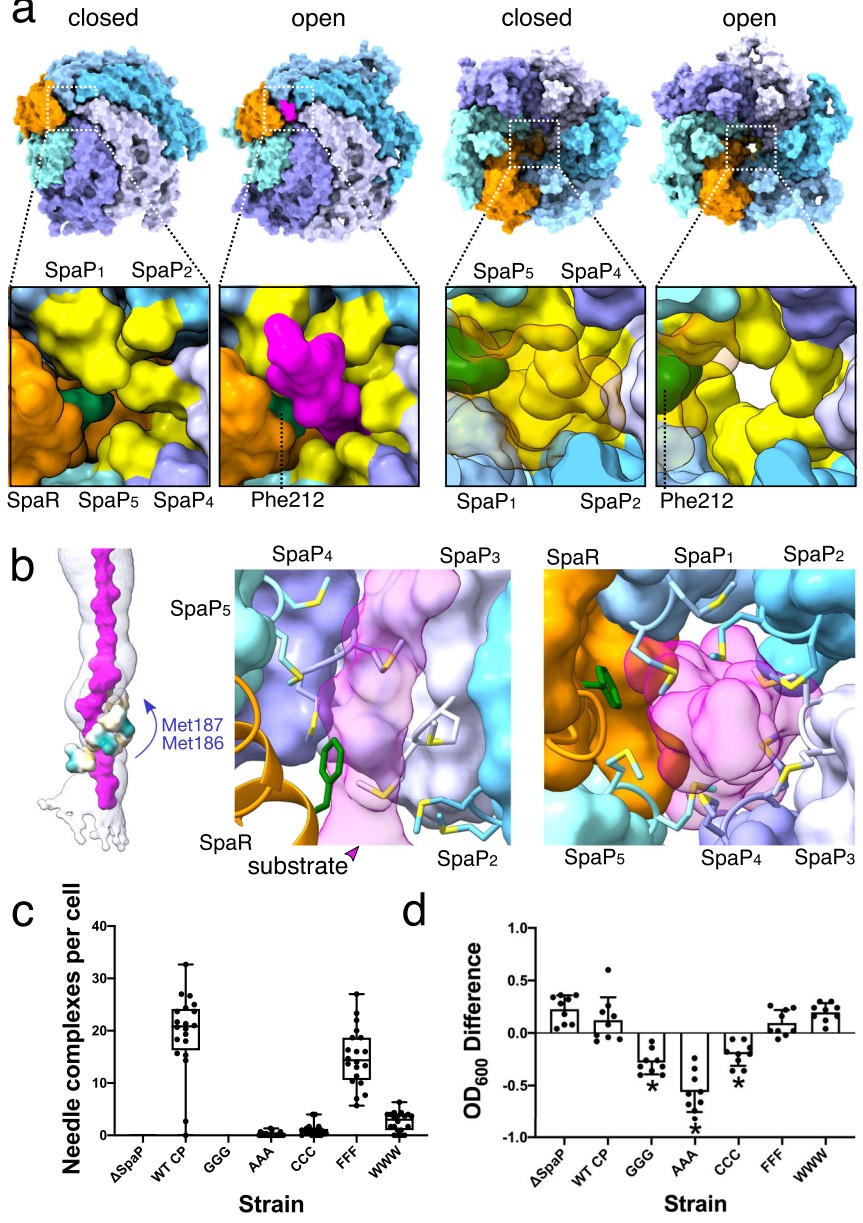

**Fig. 4 Opening of the M-gate in the EA facilitates substrate translocation and is crucial for needle complex assembly and cellular fitness. a** Bottom and top views of the EA, with M-gate shown in the lower panels, depicted in surface representation in a closed or open (substrate-engaged) state. SpaP methionine residues 186/7 are displayed in yellow and SpaR Phe212 is in green. SpaR is transparent in the top views. In the top view of the open EA (far right), the substrate has been removed to aid visualization. **b** Left: surface representations of the M-gate residues Met186/7 colored by hydrophobicity. Middle and right: side and bottom views of the M-gate staircase with Met186/7 and Phe212 depicted in stick representation. **c** Quantification of needle complexes ($n = 20$ cells) in *Salmonella* SpaP-knockout strains complemented with SpaP WT (WT CP) or with mutants targeting the conserved 185-Met-Met-Met-187 motif of the M-gate (GGG-WWW). Boxplot whiskers show min and max counts. Lines in the middle of the box represent median values and box limits are the 25th and 75th percentiles. Individual squares represent the number of needle complexes averaged from three counts per cell. Needle complexes were counted by three individuals and averaged, and data represents one of three experiments. **d** Optical densities of the *Salmonella* strains in **c**. Plotted is the mean difference in $OD_{600}$ between cultures grown for 6 h under T3SS-inducing and non-inducing conditions. $n = 9$ independent samples examined over three experiments. Individual counts are represented by dots. Error bars represent SD and asterisks represent a significant difference compared to WT CP (GGG $P = 1.33 \times 10^{-6}$, AAA $P = 2.93 \times 10^{-13}$, CCC $P = 9.31 \times 10^{-5}$). A one-way ANOVA F(6,56) = 37.42 and a Bonferroni's multiple comparisons test were used to assess statistical significance between the WT complemented control (WT CP) and M-gate substitution strains. Source data for Fig. 4c are provided with this paper.

accommodate the unfolded substrate chain. By intimately engaging the translocating substrate, the M-gate effectively acts as a tight seal to facilitate transport but maintain the physical boundaries between the pathogen's cytoplasm and (i) its periplasm during needle complex assembly, (ii) the outside environment (prior to infection), and (iii) the host cell cytoplasm (during infection).

**SpaR contains a retracting loop/lid and SpaP$_{1-5}$ form a second hydrophilic Q2-belt, together facilitating substrate transport**. In the apo-state, a unique loop of residues 106-123 of SpaR extends horizontally out on top of the M-gate (Fig. 5a and Supplementary Video 2). Consistent with the recombinant EA from *Shigella flexneri* injectisomes, SpaR residues Leu110 and Ile114 (Phe110 and Phe113 in FliP) interface with the methionines of the

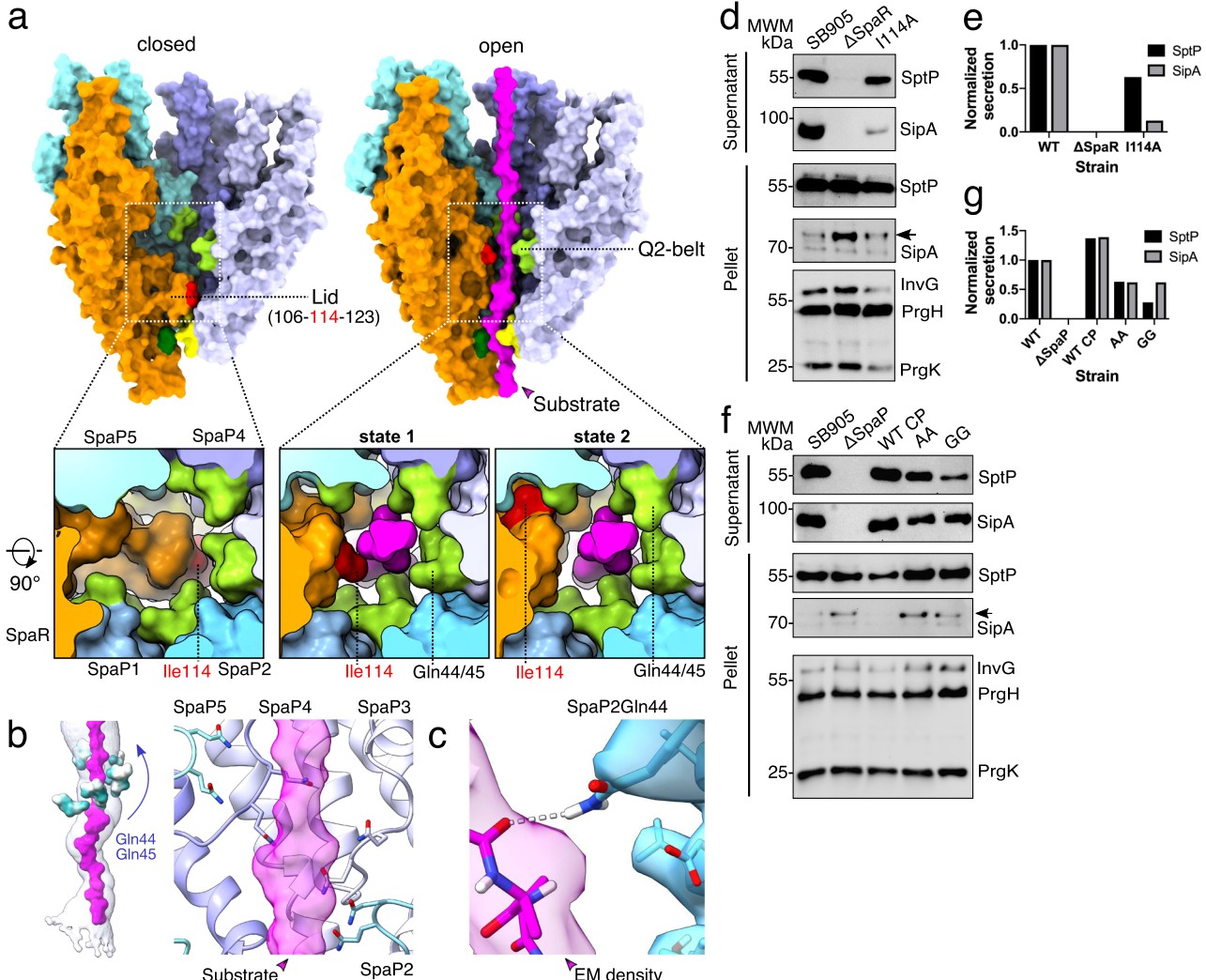

**Fig. 5 Conserved glutamines in SpaP and the SpaR lid together orchestrate substrate transport through the Q2-belt of the EA. a** Surface representations of the EA showing the SpaR loop/lid and surrounding Q2-belt. Q2-belt residues are shown in light green, M-gate residues in yellow. Numbers correspond to residues building the SpaR lid. Phe212 and Ile114 are displayed in dark green and red, respectively. **b** Left: surface representations of the Q2-belt residues Gln44/45 colored by hydrophobicity. Right: close up of Q2-belt with Gln44/45 displayed in stick representation and SpaPs as ribbon diagrams. **c** Hydrogen-bond formation between the substrate backbone and SpaP$_2$ Gln45. The surfaces represent EM density (threshold: 0.016). **d** Representative immunoblots of three independent secretion assays of SptP and SipA effector proteins using wild-type SB905, SpaR knockout (ΔSpaR), and a strain carrying the SpaR loop mutation I114A. **e** Densitometry quantification of SipA and SptP secretion assay immunoblots in **d**. **f** Representative immunoblots of three independent secretion assays of SptP and SipA effector proteins using wild-type SB905, SpaP-knockout (ΔSpaP), and ΔSpaP strains reconstituted with either wild-type SpaP (WT CP), or the indicated SpaP Q2-belt Gln44,45 mutants. **g** Densitometry quantification of SipA and SptP secretion assay immunoblots in **f**. Source data for **e**–**g** are provided with this paper.

M-gate below, creating what has been termed a plug in the structurally-related flagellar system (Supplementary Figs. 19 and 23)[12,13,15]. In the map of our substrate-engaged complex, the SpaR loop (or aptly named lid) adopts two distinct conformational states positioned vertically along the translocation path (Fig. 5a and Supplementary Fig. 23 and Supplementary Video 2). In state 1 (7ah9), the predominant state in our substrate-trapped structure, the SpaR loop generates a narrow path, ~6 Å in width, to make way for the substrate on its passage to the more spacious atrium. Stabilized by the formation of an antiparallel β-sheet, the hydrophobic side chain of SpaR Ile114 is exposed towards the channel lumen, where it directly faces the translocating substrate (Fig. 5a and Supplementary Fig. 24). Mutational analysis revealed that substitution of Ile114 to alanine impairs SptP and SipA effector protein secretion, indicating that the SpaR loop actively contributes to substrate transport (Fig. 5d, e).

In state 2 (7ahi), no secondary structure elements can be observed and SpaR Ile114 is rotated away from the channel, which increases the width of the translocation path from ~6 Å to ~10 Å (Fig. 5a). Interface analyses (Proteins, Interfaces, Structures and Assemblies (PISA)) revealed that neither of the two SpaR loop conformations forms stable interactions with residues building the translocation channel (state 1: ΔG = −4.4 kcal/mol, P = 0.86; state 2: ΔG = −4.1 kcal/mol, P = 0.85), demonstrating that the SpaR loop is a moveable element that enjoys conformational flexibility during substrate transport (Fig. 5a and Supplementary Table 2).

The area surrounding the substrate on the height of the upfolded SpaR loop is shaped by a loop which connects α-helices α2 and α3 in each of the five spirally organized SpaP protomers (Fig. 5b). Reminiscent of the Q1-belt function and conserved across virulent T3SSs, SpaP Gln44 and Gln45 localize within close

proximity to the substrate, together generating an extended hydrogen-bonding donor/acceptor interface to engage with the backbone and polar side chains of the SptP substrate as it emerges from the M-gate and passes SpaR Ile114 (Fig. 5c and Supplementary Figs. 12, 16, and 17). Because of the striking similarity with the Q1-belt, we decided to name this region of the translocation path Q2-belt. Mutation of the two SpaP glutamines Gln44 and Gln45 had only a moderate impact on SptP and SipA effector protein secretion, revealing that the Q2-belt plays an ancillary role in substrate transport, which is also in agreement with a lower conservation of these residues in the homologous flagellar protein FliP (Fig. 5f, g and Supplementary Fig. 17).

It is noteworthy that in state 2 (~10 Å) but not state 1 (~6 Å) of the SpaR loop, the Q2-belt provides sufficient space to also accommodate α-helices, suggesting that the events that cause conformational switching of the SpaR loop may not be limited to unfolded substrates but may also be caused by translocating α-helices, which may provide an additional explanation for the evident ambiguity of the substrate density in our map.

**The PrgJ inner rod is supported by lipids; PrgJ1 and PrgI$_{1,4-5}$ form unique contacts to InvG.** Besides its role as a translocon, the EA also functions as a structural scaffold onto which the inner rod protein PrgJ assembles (Figs. 1c and 2a). Six PrgJ protomers interface with $1 + 5$ SpaR and P proteins and previously unresolved lipids, together creating the atrium. This spacious chamber connects the channel, defined by the conical architecture of the EA, with the tunnel of the helical needle filament (Fig. 2). The lipids reside in a circular gap present in the upper EA, where they function to accommodate SpaP α-helix α1 and stabilize the helical fork of PrgJ, together forming a nucleation seed that drives polymerization of the needle filament (Fig. 6a). All PrgJ proteins cross the EA/InvG gap to engage into β-strand complementation interactions formed between their N-termini and β-sheet β6 of the surrounding InvG subunits together stabilizing the inner rod within the confinement of the basal body (Supplementary Fig. 25). Notably, the lowest monomer PrgJ1 has a unique fold, which extends horizontally, interfacing the SpaR loop connecting α-helices α2 and α3 before traversing SpaP1 to then cross the gap between the EA and the basal body, clarifying earlier reports in which this part could not be resolved and PrgJ2 was speculated to interface in alternate locations (Supplementary Fig. 25)[6]. Likewise, also N-termini of previously unresolved PrgI$_{1,4-5}$ protomers cross the EA/InvG gap to interact with SpaP, PrgJ, and the basal body component InvG, with each of these PrgI N-termini forming unique, plastic interactions with its respective environment, providing a rationale as to why some mutations localizing to the PrgI N-terminus abrogate filament assembly in cellulo but not in vitro (Supplementary Fig. 26)[24].

**Substrates may contain α-helices during transport through the atrium and needle filament.** The needle complex structure presented here provides a refined view on the helical PrgI filament which was built de novo into our non-symmetrized C1 map and hence, unlike in all other structures, no helical symmetry or restraints have been imposed. With an average axial helical rise of 4.41 Å between subunits and a pitch of ~5.5 subunits, our C1 filament grows ~24.3 Å per turn compared to ~23.8 Å (6dwb), ~23.3 Å (6ofh), and ~23.1 Å (2lpz) in models obtained by helical reconstruction cryo-EM and nuclear magnetic resonance (NMR) spectroscopy, respectively[6,7,24]. The quality of our map allowed us to model 72 PrgI subunits, covering a distance of ~36 nm and therefore our filament accumulates a total size difference of at least ~5.5 Å. Despite this difference, the filament tunnel of our substrate-engaged structure adopts the form of a right-handed

helix with a minimal inner diameter of ~13 Å, which is indistinguishable from published apo-state structures and sufficiently large to accommodate α-helices (Fig. 6b)[7,8,14,24]. At the passage from the Q2-belt to the atrium, the density corresponding to the substrate diminishes, demonstrating that, in contrast to the three tight interfaces seen in the EA, the wider lumen in the atrium (~13.5 Å vs. ~5.5–10 Å) provides the substrate with higher conformational flexibility (Figs. 2 and 6b,c, and Supplementary Fig. 6). Interestingly, the substrate density reappears in the upper atrium from where it continues through the tunnel of the filament (Fig. 6c). Here, it assumes a tubular shape which, especially at higher map thresholds, is remarkably similar to those of α-helices at low-resolution, indicating that substrates that enter the secretion system potentially retain secondary structure elements, or alternatively, may partly refold during their passage through the filament (Fig. 6c).

## Discussion

Here we report a high-resolution snapshot of a virulent type III secretion system in an active, substrate-engaged conformation, providing insight into the molecular basis of protein transport across the bacterial envelope, a process that is fundamental to the virulence of many pathogenic bacteria and the motility of all flagellated bacteria. Our substrate-engaged structure reveals the complete secretion channel through the EA core complex, confirming its role as an entry portal to the needle complex.

Surprisingly, the EA lumen exhibits most of the conformational changes seen in the substrate-engaged structure, an unexpected finding given the sheer size and complexity of the needle complex machine. This agrees with, albeit at lower resolutions, our earlier structure and with visualizations of in situ needle complexes contacting host cells, together suggesting that the needle complex forms a largely static channel, in contrast to other more dynamic secretion machines[9,18,25]. In fact, many residues involved in substrate translocation localize to loop regions, which together with low rmsd values between apo-state and substrate-engaged structures supports the concept that the basal body rings and the bulk of the SpaPQR complex provide a scaffold to position critical residues in the secretion channel. It appears plausible that this rigid architecture is a necessity to traverse the bacterial envelope, to provide a stable docking base for the dynamic components of the cytoplasmic sorting platform and simply to withstand the forces two moving cells and also the translocation process itself, exert on the secretion system[4,26,27].

In line with this concept, all six PrgJ proteins of the inner rod and PrgI$_{1,4}$ of the filament base engage into interactions with the surrounding InvG proteins, together stabilizing the translocation conduit within the confinement of the basal body. To connect the conical shape of the EA with the helical filament, the inner rod protein PrgJ assumes a fold of a helical fork that is similar to those of the filament protein PrgI (Supplementary Figs. 25 and 26). Consequently, lipids that we find determine the shape of PrgJ, and hence the entire rod, are vital to its function as a nucleation seed that facilitates needle polymerization.

Strikingly, our substrate-trapped structure confirms the essential role of the EA core complex in T3 secretion, forming a conserved three-point interface with the substrate. The EA channel contains two hydrophilic Q-belts sandwiching the hydrophobic M-gate and movable SpaR lid, which together function as a gate and guide, to engage and steer substrates to the filament. By establishing complementary hydrophilic interactions with the substrate backbone, conserved glutamines can facilitate effector protein translocation in a side chain-independent manner, an intriguing possibility due to the variety of different substrates accepted by injectisomes, which have also been exploited

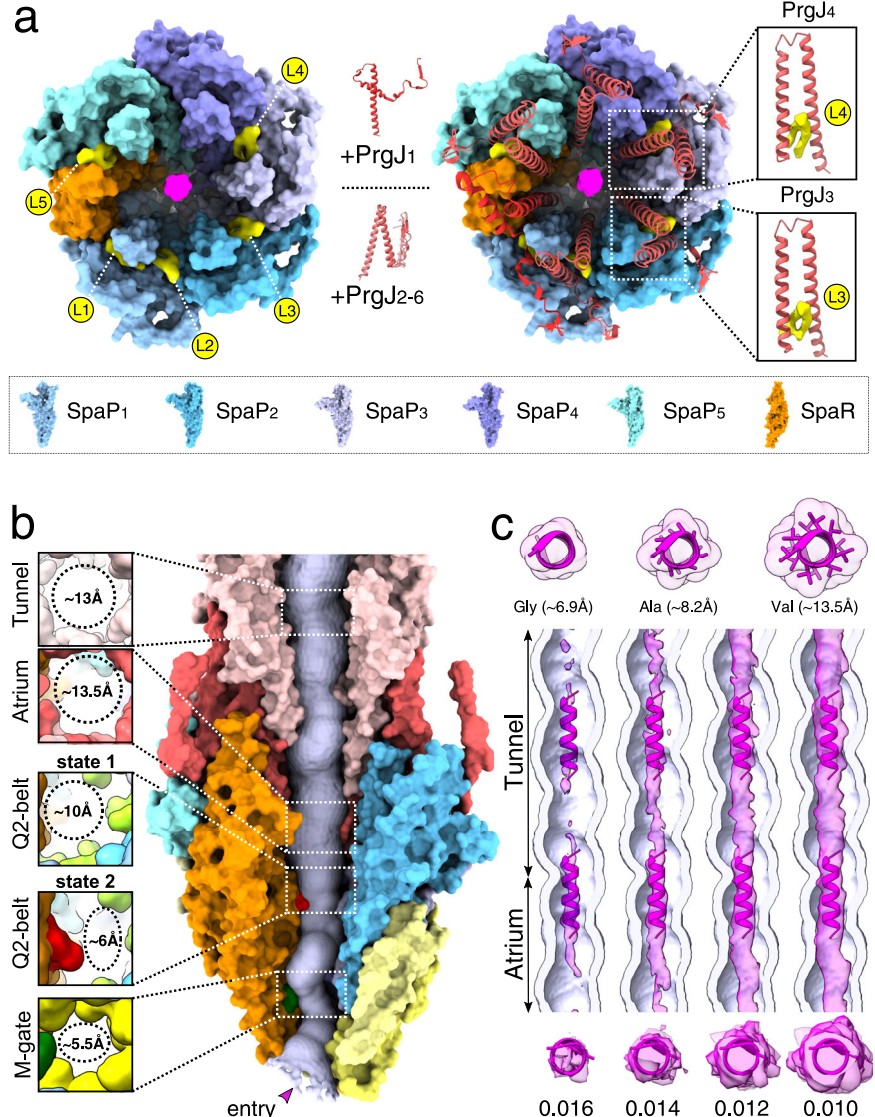

**Fig. 6 Density in the lipid-stabilized atrium and the filament tunnel suggests possible transport of partially-folded substrates. a** Top views of EM densities corresponding to lipids (yellow, L1–L5) residing in a circular gap formed by SpaP and SpaR proteins in the EA. The lipids stabilize PrgJ α-helices α-1 and α-2 forming the PrgJ forks (right boxes; PrgJs displayed as red ribbon diagrams). **b** Size constraints of the substrate translocation path through the needle complex. M-gate (⌀ ~5.5 Å), Q2-belt (⌀ ~6 Å, state 1; ~10 Å, state 2), atrium (⌀ ~13.5 Å), tunnel (⌀ ~13 Å). **c** Top: example α-helices and surface diameter calculations of poly-Gly, poly-Ala, and poly-Val α-helices. Middle: vertical cross sections through the filament tunnel with density corresponding to the SptP3x-GFP substrate shown at increasing map thresholds. Bottom: secondary structure elements illustrate that the substrate EM density at high thresholds is sufficiently bulky to accommodate α-helices.

biotechnologically for secretion of artificially-designed substrates (e.g., nanobodies, DARPins, monobodies, spider silk monomers, and viral epitopes)[28–30].

Our data confirms the M-gate forms the main constriction in apo-state structures that shifts just enough to allow substrate passage while likely preventing unwanted transport of other molecules. Mutations that increase M-gate size impede cellular growth and injectisome secretion, which we believe is a result of creating leaky pores in the inner membrane, that likely hinder the proton motive force believed to power T3SSs[31]. Consistent with this hypothesis, deletion of one of the M-gate methionines or mutation to less bulky residues in the flagellar homolog FliP increases ion conductance, while mutation to bulkier phenylalanine residues does not[23]. Taken together, this suggests that the M-gate methionines are critical residues providing flexibility for substrate transport while forming a selective barrier[23].

The high resolution achieved here enabled us to decipher a conformational switch of the SpaR lid which opens up to then support effector protein secretion towards the filament. In the apo-state, the conformation of the SpaR loop can be traced only with sufficiently smoothed maps and at high map thresholds in both: isolated needle complexes (our structure and 6pep) and in the recombinant EA from *S. flexneri* (6r6b), suggesting that, in contrast to its structural counterpart in the flagellar T3SS (FliP residues: 105–124), the SpaR loop is only weakly associated with the M-gate located below (Supplementary Fig. 19)[12,13,15]. Together, our structural and biochemical data suggests that the Q1-belt, M-gate, Q2-belt and the SpaR lid cooperate to facilitate substrate transport, providing a rationale as to why mutation of a single interface is not sufficient to completely prohibit effector protein secretion. The SpaQs securely support the above SpaP subunits and SpaR, creating the tight gasket in the channel needed for maintaining membrane homeostasis as suggested in the

flagellar EA[12]. In the light of these concepts, it appears even more remarkable that secretion of substrates that have passed the Q1-belt and M-gate can still be compromised by a single I114A point mutation in the SpaR loop.

Our substrate-engaged structure reveals continuous density throughout the needle complex conduit, corresponding to mostly non-folded SptP3x-GFP sequences in the EA channel. In fact, the N-termini of many substrates are heterogenous in sequence and predicted to be intrinsically disordered over a length of ~15–35 residues and therefore are sufficiently long to consecutively penetrate through all three constrictions, which we envision could be the most important unifying feature of T3SS signal sequences[20]. In favor of this concept, chaperones rather than N-terminal signal sequences were found to be crucial for routing effector proteins to the injectisome sorting platform[4]. Conceivably, it appears tempting to assume that effector proteins experience a selection pressure that drives the evolution of their N-termini towards non-folding sequences to facilitate their loading into the needle complex and hence ensure secretion.

However, SicP-chaperone proteins maintain the chaperone-binding domain of SptP in a non-globular state containing α-helices, which raises the intriguing question whether or not further unfolding prior to entry into the secretion system is required[3]. Evidently, for α-helices to be directly translocated through the EA channel, the narrow, ~5.5 Å-wide lumen of the M-gate would have to open further (Fig. 6b, c). Although we cannot rule out that α-helices are in fact directly translocated through the EA, the tubular appearance of the density corresponding to the substrate in the filament in our map suggests that α-helices may fold during their passage through the needle complex. Either way, the transport of α-helices is likely advantageous, accelerating the folding of effector proteins upon arrival in the host cell cytoplasm and allowing them to elicit their effector functions faster, a process that would benefit pathogen fitness. Also, helical filament assembly could be more effectively achieved by docking of the readily formed helices into their cognate interfaces at the distal end of the growing filament. This concept is in agreement with measurements revealing that the hook length control protein FliK is being transported through the flagellar T3SS in an α-helical form[32].

Future studies will also have to address the question whether the constrictions inside the EA channel are passive components that only change their conformation as a consequence of the penetrating substrate or, alternatively, are regulated by active gating mechanisms. Given that hardly any conformational differences are seen in the scaffolding parts of the SpaPQR proteins (Supplementary Fig. 9), it is difficult to imagine a cytoplasmic signal translating up to the M-gate or SpaR lid prior to substrate engagement, instead suggesting that these structural elements shift in response to the substrate physically traveling through the channel. However, we cannot rule out the possibility that gating of the EA portal occurs through SpaS, which is mostly absent in our complexes but likely orchestrates the shape of the SpaQ loops building the Q1-belt in native injectisomes. Alternatively, SpaS could represent a physical link providing guidance for the approaching disordered N-terminus to help finding its way into the narrow cytoplasmic opening of the EA portal.

Based on the structure and biochemical work presented here, we propose a model of substrate secretion through the T3SS EA (Fig. 7). (1) After chaperone removal and substrate unfolding, the substrate is guided to the EA portal engaging with the Q1-belt, which is capable of accommodating effector proteins independent from their sequence. (2) The substrate transports up through the methionine network, disrupting its hydrophobic interface causing the methionines to shift apart and open the M-gate. (3) The SpaR loop, resting above the M-gate and blocking the channel, extends upwards, assuming an open conformation, which together with the surrounding Q2-belt creates an interface to engage the substrate. (4) The upfolded SpaR loop contacts the passing substrate, and together with the Q2-belt guides the substrate up through the atrium and into the lumen of the filament.

## Methods

**Bacterial strains and plasmids.** Experiments were conducted using a *S. enterica* sv. Typhimurium non-flagellated strain SB905 carrying the T3SS transcriptional regulator gene *hilA* expressed on the pSB3291 plasmid using an *araBAD* promoter (Supplementary Tables 3 and 4)[18]. The SptP-based substrate and the SptP chaperone SicP were both expressed using a pACYCDuet-1 CmR plasmid (Merck Chemicals). The substrate construct consists of an N-terminal signal sequence, a SicP-chaperone-binding domain, three SptP effector domain repeats fused to enhanced GFP, and a 3× FLAG tag[18].

**Genome engineering.** *Salmonella* strains with genomic mutations targeting the SpaQ Q1-belt, SpaR M-gate, and the SpaR loop were generated using the pORT-MAGE genomic engineering workflow[33]. Briefly, SB905 carrying a pORTMAGE3-KanR plasmid was grown for 30 °C and then at 42 °C for 15 min, made electrocompetent, and electroporated with oligos carrying the specific mutations targeting SpaQ or SpaR. Subsequent transformation cycles were repeated until positive clones were found by gene sequencing. The pORTMAGE plasmid was cured by transformation with plasmids carrying *hilA*. *spaP*, *spaQ*, and *spaR* gene knockout strains were generated by recombineering methods[34]. Briefly, a CmR cassette with flanking FLP flp recognition target sites, was used to knockout the EA genes of interest in a *Salmonella* strain containing a lambda Red recombinase. Subsequently, the genomic region was transferred to a clean SB905 background by P22 phages. The resistance gene was removed by flipping it out with a pCP20 plasmid containing a yeast FLP recombinase[34].

**Bacterial growth assays.** Overnight cultures of *Salmonella* SB905 SpaP-knockout strains complemented with either WT SpaP or with GGG, AAA, CCC, WWW, and FFF substitutions were grown with or without 0.3 M NaCl supplementation (injectisome inducing vs non-inducing conditions) under antibiotic selection. The following morning, cultures were normalized to an $OD_{600}$ of 0.5 before diluting 1:10 in either non-inducing lysogeny broth (LB) or inducing media, LB supplemented with 0.3 M NaCl, and 0.012% w/v arabinose, for injectisome assembly. Cultures were sampled at 1 h intervals and $OD_{600}$ measurements were recorded using a spectrophotometer to calculate growth curves. Three independent experiments of three biological replicates were used for each strain and condition. Data and statistical analyses were visualized using the GraphPad Prism8 software package.

**Secretion assays.** Overnight cultures of *Salmonella* SB905, the SpaQ and SpaR mutants or the complemented SpaP-knockout strains were grown at 37 °C in 5 ml of LB medium containing 0.3 M NaCl and antibiotics. The next morning, cultures were diluted 1:10 in 50 ml of the same medium without antibiotics. The expression of HilA was induced by addition of 0.012% w/v L-arabinose for another growth period of 5 h. Afterwards, the cell density was adjusted to an $OD_{600}$ of 1.0 and the cells were pelleted at $3146 \times g$ for 10 min. Supernatants were collected and filtered with a 0.22 μM syringe filter. Pellets were resuspended in 1× phosphate-buffered saline (PBS). Both supernatants and cell pellets were then immunoblotted with antibodies raised against effectors SptP and SipA or needle complex proteins (generated in-house, 1:5000 dilution for anti-SptP and 1:10,000 dilution for anti-SipA and anti-needle complex), and then secondary anti-rabbit or anti-mouse horseradish peroxidase (HRP)-conjugated antibodies (Sigma Aldrich/Merck kGaA; 1:10,000 dilution each). Immunoblots were imaged using Intas Science Imaging Instruments GmbH ChemoStar Touch imager running v.0.5.65 software and were quantified using ImageJ v.1.50i software[35].

**Needle complex counting.** Overnight cultures of *Salmonella* SB905 SpaP-knockout strains complemented with either WT SpaP or with M-gate substitutions were grown with 0.3 M NaCl supplementation under antibiotic selection. The following morning, cultures were diluted 1:10 in injectisome-inducing media, LB supplemented with 0.3 M NaCl, 0.012% w/v arabinose and without antibiotics. After 5 h of growth, cultures were diluted to an $OD_{600}$ of 1.0 and 1 ml was collected and centrifuged for 5 min at room temperature (RT), ~16,000 × g. The pellets were resuspended in ~30 μl of LB and incubated at RT for 5 min. 1 ml of ice-cold water was mixed with the cells, which were then incubated on ice for 5 min. DNAse (1.5 μl, Thermo Fisher Scientific) was added along with 20 μl of 1 M MgCl₂ and incubated at RT for 10 min. Forty microliters of 0.5 M EDTA and 150 μl of 1 M Tris-Cl (pH 7.5) were added and the cells were centrifuged for 30 min at 4 °C, ~16,000 × g. Pellets were resuspended with 1 ml of ice-cold Tris-EDTA (TE) buffer and centrifuged again. Pellets were then resuspended in 10–50 μl of ice-cold TE buffer and incubated at 4 °C for ~20 min while shaking. Cells were applied to transmission electron microscope (TEM) grids and stained (see below) before

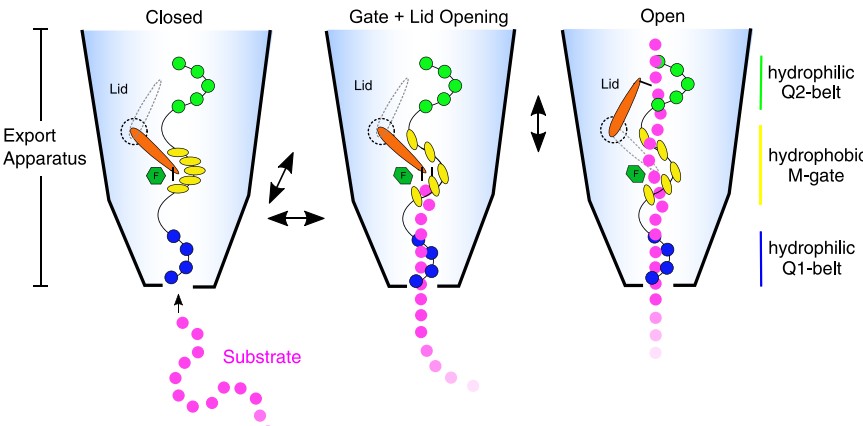

**Fig. 7 Model for substrate translocation through the EA of T3SSs.** Disordered N-termini of effector proteins (magenta circles) enter the Q1-belt (blue circles) of the EA facing the bacterial cytoplasm. Successive binding of the substrate backbone to conserved glutamines facilitates penetration through the Q1-belt and loading of the substrate into the needle complex. The substrate then penetrates the M-gate. Disruption of its hydrophobic interface (Met186-Met187: yellow ovals; SpaR Phe212: green hexagon) expands the gate and causes the SpaR loop (orange) to flip into a vertical position together generating a narrow path for the translocation of the unfolded substrate chain. The substrate then proceeds up to the Q2-belt (green circles) and SpaR loop (orange; Ile114 represented as black tick), which together stabilize the substrate above the M-gate and support substrate transport towards the atrium and needle filament to ultimately facilitate effector protein secretion.

imaging with a Thermo Fisher Talos L120C TEM. Twenty cells were imaged at varying magnifications and needle complexes were counted by three colleagues, independently. Counts from each person for each cell were averaged and the data was visualized using the GraphPad Prism8 software package.

**Purification of needle complexes**. Needle complexes were purified either from *Salmonella* SB905 WT or SpaP KO strains. Briefly, day cultures were grown for 5 h in LB media supplemented with 0.3 M NaCl, 0.012% w/v arabinose and without antibiotics prior to harvesting. Cells were then pelleted at $4250 \times g$ for 12 min at 4 °C and resuspended in 0.5 M sucrose, 0.15 M Tris pH 8.0 buffer. Resuspensions were transferred to bottles on ice and mixed continuously with a magnetic stirrer. 12.5 mM EDTA and 30 mg/ml of lysozyme were added to the suspensions which were then kept stirring for 45 min. Tablets of cOmplete™ mini EDTA-free protease inhibitor cocktail (Sigma Aldrich/Merck KGaA) were added and the cell suspensions were transferred to 37 °C water baths and 0.4% w/v N,N-Dimethyl-dodecylamine N-oxide (LDAO) followed by 0.45 M NaCl and 20 mM MgCl₂ were added and mixed. Cell debris were pelleted at $14,000 \times g$ for 20 min and membranes were pelleted in a Beckman Coulter type 45-Ti rotor at $186,000 \times g$ for 2.5 h at 4 °C. Membranes were resuspended in PR2 buffer (10 mM Tris, 0.5 M NaCl, 5 mM EDTA 0.5 % w/v LDAO pH 8.0) and afterwards CsCl and LDAO were added to a final concentration of 26.25% w/v CsCl, 0.5% w/v LDAO. Needle complexes were separated by centrifugation overnight in a Beckman Coulter SW 60 Ti rotor at $336,800 \times g$ at 4 °C. Fractions containing assembled needle complexes were pelleted in a Beckman Coulter TLA-110 rotor at $439,800 \times g$ for 35 min at 4 °C, resuspended in FR3 buffer (10 mM Tris-HCl pH 8.0, 0.5 M NaCl, 5 mM EDTA, 0.1% w/v LDAO) and used for negative staining or cryo-EM. For purification of substrate-trapped complexes, SB905 expressing HilA and the substrate SptP3x-GFP with the chaperone SicP was grown for 4 h with an additional 2 h induction period using 1 mM isopropyl β-D-1-thiogalactopyranoside (IPTG) prior to harvesting. To enrich for substrate-trapped complexes, purified needle complexes were pooled and incubated with anti-FLAG M2 magnetic beads (M8823, Sigma Aldrich/Merck KGaA) prewashed in FR3 buffer for 2.5 h under gentle agitation at 4 °C. The beads were subsequently washed 12 times with FR3 buffer for 10 min each. Beads were eluted with FR3 buffer twice as mock elutions and subsequently eluted twice for 45 min with 2 mg/ml 3× FLAG peptide in FR3 buffer. These elutions were then pooled and pelleted in a Beckman Coulter TLA-110 rotor at $439,800 \times g$ for 35 min at 4 °C. Needle complex pellets were resuspended for at least 1 h in FR3 buffer under agitation at 4 °C before TEM imaging and SDS-polyacrylamide gel electrophoresis. Immunoblots were probed with anti-needle complex or anti-FLAG antibodies (1:10,000 dilution each; anti-FLAG antibody from Sigma Aldrich/Merck kGaA) and then secondary anti-rabbit or anti-mouse HRP-conjugated antibodies (Sigma Aldrich/Merck kGaA; 1:10,000 dilution).

**Negative staining TEM**. Four microliters of diluted sample was applied to carbon-coated copper grids and incubated for 40 s. Grids were glow discharged before for 30 s at 25 mA using a GloQube® Plus Glow Discharge System (Electron Microscopy Sciences). The sample was blotted off and the grid was washed briefly with 4 μl of staining solution (2% w/v phosphotungstic acid (PTA), adjusted to pH 7.0 with NaOH) and then stained with 4 μl of the staining solution for 20 s. The stain was blotted off and the grids were air-dried for at least 1 min. Grids were imaged

using a Thermo Fisher Scientific Talos L120C TEM with a 4 K Ceta CEMOS camera using Talos v.1.15.3 and TEM Imaging and analysis v.5.0 software.

**Sample vitrification and cryo-EM data collection**. Purified apo-state needle complexes were applied to Quantifoil grids with either an additional layer of amorphous carbon (<1.6 nm thick) or graphene oxide. Purified, substrate-trapped needle complexes were applied to Quantifoil grids floated with an ~1.1 nm layer of amorphous carbon on top. Four microliters of sample was applied onto glow-discharged grids (30 s, 25 mA) and allowed to disperse for 0.5–2 min. The grids were blotted for 4–7 s set at 100% humidity and plunge frozen in a liquid propane/ethane mixture cooled to liquid nitrogen temperatures, using a Thermo Fisher Scientific Vitrobot Mark V. Vitrified samples were imaged on a Thermo Fisher Scientific Titan Krios TEM operating at 300 kV and equipped with a field emission gun (XFEG) and a Gatan Bioquantum energy filter. Movies consisting of 25 (apo) or 50 (substrate-trapped) frames, were automatically recorded using Thermo Fisher Scientific EPU software v. 1 or 2, and a Gatan K2 or K3 camera, at 0.3–5.2 μm defocus in counting mode (Table 1).

**Single particle analysis**. Single particle analysis was performed using Relion 3.0 for the apo-state needle complex and Relion 3.1-beta for the substrate-trapped needle complex[36,37]. Movies were motion-corrected using MotionCor2 v.1.2.1 and v.1.3, dose-weighted, and the contrast transfer function (CTF) was determined using CTFFIND4 v.4.1.13[38,39]. Coordinates were determined from the motion-corrected micrographs using crYOLO v.1.2 and v.1.4[40] trained with a subset of manually picked particles. Particles were extracted and binned for several rounds of two-dimensional classification. A cleaned and unbinned data set was obtained by re-extraction and aligned to a rotationally averaged structure. Focused refinements with and without applying symmetry were performed to the individual sub-structures using respective 3D masks. After converged refinements, per particle CTF and Bayesian polishing were used to generate a new data set for another round of focused refinements. Final rounds of refinement were performed without any masking of particles and symmetrization. Overall gold-standard resolution (Fourier shell correlation = 0.143) and local resolution, as well as sharpened maps (B-factor: −30), were calculated with Relion 3.1-beta[37].

**Model building, refinement, and validation**
*Apo-state*. Model building into the T3SS apo-state map started by placing homology models for SpaP, SpaQ, and SpaR, which were generated with the SWISS-MODEL server using *V. mimicus* FliP, FliQ, and FliR structures as templates (6s3l); PrgJ was modeled using Phyre2 v.2[15,41,42]. Together with PrgI (2lpz), homology models for SpaP, SpaQ, SpaR and PrgJ were first fitted into the EM map using the fit-in-map tool in UCSF Chimera (v.1.14) and then manually extended with Coot (v0.9-beta)[7,43,44]. A first refinement was performed with Rosetta v.3.1 controlled via StarMap v.1.1.12 (manuscript in preparation), followed by inter-active refinement against the map density with ISOLDE v.1.0b5, a molecular dynamics-guided structure refinement tool within ChimeraX v.0.93[45,46]. The resulting coordinate file was further refined with Phenix.real_space_refine v.1.18-6831 using reference model restraints, strict rotamer matching and disabled grid

search[47]. Model validation was carried out using MolProbity server and EMRinger v.1.00 within the Phenix software package (Table 1)[48–50].

*Substrate-trapped state.* Existing models of PrgH (3gr1), PrgK (3gr5), InvG (4g08, G34-I173: lower OR), SpaP, SpaR, SpaS, PrgJ, and PrgI were rigid body-fitted into the electron density map using the fit-in-map tool in UCSF Chimera v1.14, followed by manual rebuilding in Coot v0.9-beta[43,44,51,52]. The upper OR (InvG: E174-G557) was first built into a C15-symmetrized and focus-refined map using Coot (v0.9-beta) for ab initio model building, followed by rigid body-fitting into the C1 density map using the fit-in-map tool in UCSF Chimera v1.14[43,44]. Interactive refinement against the C1 map density was performed with ISOLDE v.1.0b5[45]. The resulting coordinate file was further refined with Phenix.real_space_refine v.1.18-6831 using reference model restraints, strict rotamer matching and disabled grid search[47]. Model validation was carried out using MolProbity server and EMRinger v.1.00 (Table 1)[48–50]. The translocation channel through the EA and lumen of the needle filament was calculated using HOLE v.2.2[53]. Putative pore sizes of SpaP$^{CCC}$, SpaP$^{AAA}$ and SpaP$^{GGG}$-containing EAs were determined using the mark region tool in combination with the distance measurement tool in ChimeraX[46]. The helical rise of the PrgI filaments from the substrate-trapped filament, 2lpz, 6ofh, and 6dwb were measured by running the Rosetta tool make_symmdef_file.pl for each consecutive pair of PrgI protomers in the direction from base to tip[7,8,24,54]. Measurements were limited to 26 protomers and started from PrgI$_{12}$ in the substrate-trapped filament. UCSF Chimera and ChimeraX were used for molecular visualization[43,46].

**Reporting summary**. Further information on research design is available in the Nature Research Reporting Summary linked to this article.

## Data availability

Maps have been deposited at the EMDB with accession codes EMD-11780 and EMD-11781. Models have been deposited at the PDB with accession codes PDB 7agx, 7ah9, and 7ahi. All other datasets generated and/or analyzed and materials used during the current study are available from the corresponding author on reasonable request. Source data are provided with this paper.

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

## Acknowledgements
We thank all members of the Marlovits Laboratory, including Catalin Bunduc, Biao Yuan, and Barbara Grueter, for their support of this project. We thank Wolfgang Lugmayr and Frank DiMaio for their help with StarMap, Rosetta, and the filament analysis. We also thank Tristan Croll for his significant support with ISOLDE. High-performance computing was possible through access to the HPC at DESY/Hamburg (Germany) and the Vienna Scientific Cluster (Austria). Part of this work was performed at the CryoEM Facility at CSSB (Hamburg) and the VBCF (Vienna).

## Author contributions
S.M., D.F., and T.C.M. designed experiments. S.M.and D.F. generated constructs. N.G.M. and M.P. generated knockout strains. S.M., V.K., and O.V. purified complexes. S.M., M.P., and D.F. performed biochemical assays. S.M. and J.W. vitrified samples and collected cryo-EM images. S.M. collected negative stain images. S.M., D.F., and N.G.M. built the atomic model. S.M., D.F., N.G.M., and T.C.M. interpreted data. T.C.M. processed cryo-EM data. S.M., D.F., and T.C.M. wrote and revised the paper. All authors read, corrected, and approved the manuscript. T.C.M. conceived the study and supervised the project.

## Funding
This project was supported by funds available to T.C.M. through the Behörde für Wissenschaft, Forschung und Gleichstellung of the city of Hamburg at the Institute of Structural and Systems Biology at the University Medical Center Hamburg-Eppendorf, the Institute of Molecular Biotechnology (IMBA) of the Austrian Academy of Sciences, and the Research Institute of Molecular Pathology (IMP). T.C.M. (and S.M.) received funding through grant I 2408-B22 furnished by the Austrian Science Fund (FWF). D.F. was funded by a DFG research fellowship return grant (FA1518/2-1). V.K. was supported by Boehringer Ingelheim Fonds PhD fellowship. The CryoEM Facility at CSSB is supported by the UHH, the UKE, and DFG grant numbers (INST152/772-1 | 152/774-1 | 152/775-1 | 152/776-1 | 152/777-1 FUGG). Open Access funding enabled and organized by Projekt DEAL.

## Competing interests
The authors declare no competing interests.
