## [Peer Review File · Nature Communications]

REVIEWER COMMENTS

Reviewer #1 (Remarks to the Author):

This manuscript is well-written by Miltic et al. and presents substrate-engaged T3SS structures revealed by cryo-EM single-particle analysis to shed light on its gating mechanism for unfolded protein translocation. Two huge complex structures, apo and substrate-loaded, have been determined at enough high resolution to identify residues which involve in substrate recognition in the export apparatus (EA). In the substrate-loaded structure, the inner conduit of T3SS was occluded by an engineered effector protein that had been trapped functionally inside EA. The authors demonstrate that these structures allow us to understand a board substrate specificity in terms of the M-gate.

With proper labeling of residues in the figures, it would be helpful for general readers to gain a much better understanding of the content.

Comments:

p.2 line 10;

This should not be in abbreviation. Here use "Serovar" instead of "sv".

p.5 line 23;

Which is proper "Focussed or Focused"? "Focused" is used in p20. line 22 and 24.

p.6 line 4;

"Protein Data Bank Accession Code" should be included at first appearance in the main text.

p.6 line 13;

Please indicate four residues included in this loop region and show labels of these residues in the figure 3 a and 3 c.

p.6 line 14;

Please show a label of the corresponding residue in the figure 3 a and 3c. Clearly indicate each residues with labels.

p.6 line 23;

Does "bacterial" mean flagellar and virulent T3SSs? If this is the case, are these Gln residues conserved in FliQ of flagellar T3SSs?

p.7 line 19-21;

Please include residue IDs of these conserved methionines in the main text.

p. 8 line 7;

Please include labels for the corresponding Met residues in the Fig. 4.

p.14 line 16 and 17-18;

Is "T3SS secretion" used properly?

p.15 line 3;

Please include residue IDs for "phenylalanines".

p.17 line 16; p19 line 4;

Please correct "sicP" to "SicP".

Reviewer #2 (Remarks to the Author):

I am writing with great surprise to be reviewing the manuscript of Miletic et al ("Substrate-engaged type III secretion system structures reveal gating mechanism for unfolded protein translocation") for Nature Communications. This tour de force of structural biology, biochemistry, membrane biophysics establishes for the first time at the molecular level, with an unprecedented level of detail and mechanistic explanation, the model of non-globular polypeptide translocation through the protein Type Three Secretion Systems, with a fantastic imaging of a substrate in the process. It is the capstone on three decades of research in this area of bacterial pathogenesis, presents some astonishing EM work, and will be a powerful contribution to protein translocation in general. The work is very solid and impressively illustrated and explained. I am amazed that this is not in Nature itself as an article, especially considering the frankly underwhelming but "hot topic" manuscripts that so often pass muster for the top tiered journals. While there are minor points to quibble with, I'll let the other reviewers present such critiques. I will simply leave with the assertion that this should easily be published in Nature Communications. I will go further in fact and strongly recommend to the head editors that they go lobby to the Nature editors that this excellent and groundbreaking work be given the visibility it deserves in that journal. It would be a travesty not to do so.

Reviewer #3 (Remarks to the Author):

In the present manuscript, Miletic et al. report the first high-resolution structure of a type-III secretion system (T3SS) in an active, substrate-engaged conformation. The authors further validate that a conserved methionine-rich loop at the base of the export apparatus forms a gasket during substrate protein transport in order to maintain the membrane barrier function.

The manuscript is well written and the structure of a substrate-engaged T3SS constitutes a major advance for our understanding of protein secretion via T3SS. Accordingly, the present report is well-suited for the broad readership of Nature Communications.

The experiments concerning the proposed gating mechanism largely support the author's conclusions, but some concerns remain. I have the following comments and suggestions to improve the author's conclusions and the clarity of the manuscript.

Comments:

The potential roles of the Q-belts and R-lid are the weakest part of the manuscript and remain largely speculative. Further functional evidence would be needed to go beyond stating mere hypotheses.

In particular, concerning the SpaQ Q1-belt, only the first Q (Q41) is conserved in the flagellar T3SS (Q43), which raises questions concerning the significance of the proposed Q-belt. Similarly, the Q44Q45 residues in the SpaP Q2-belt are not conserved in FliP.

It should be shown if e.g. Q-belt mutations to alanine prevent substrate translocation to support the claim that the Q-belt has a physiological role in engaging the substrate in the channel.

Concerning the potential significance of the R-lid. The authors claim that I114 could act as the pawl of a ratchet (i.e., preventing substrates to go backwards), which should be substantiated by experimental validation. For instance, if the SpaR I114 functions a pawl that prevents substrate backward translocation, does a I114A/G mutation decrease immunoprecipitation efficiency?

Another major concern is the effect plasmid-overproduction of SpaP point mutants (Figure 4, Supplementary Fig. 18). Does episomally expressed SpaP assemble properly in the EA complex with the correct stoichiometry?

I would assume that SpaQ and SpaR, which are still under their native physiological expression control might not be present in the corresponding levels?

Accordingly, any growth differences would be much more convincing if the SpaP mutants would be expressed under native, physiological conditions.

Further, significant growth differences between the vector control, the WT SpaP and the SpaP point mutants are unclear as also the vector control shows different growth in the presence or absence of inducer?

Additional comments:

- Page 8-9: It has already been shown previously for FliP, the SpaP homolog of the flagellar T3SS that the M-gate_AAA mutation, but not the M-gate_FFF mutation induces a strong membrane permeability (Ward et al. Mol Microbiol, 107: 94–103). This should be discussed.

- Page 15: What is the evidence that the flagellar EA rotates, hence necessitating the conserved phenylalanine residues in FliR? Even if the flagellar EA rotates, it is unclear if centrifugal forces play a role at these size scales?

- Page 15: Evidence from the flagellar T3SS suggests that substrate proteins are likely secreted in an alpha-helical conformation (e.g. as shown for the hook length control protein FliK; Shibata et al. Mol Microbiol, 64: 1404–1415).

- Figure 4: Why are the leaky M-gate mutants impaired in substrate protein transport? Perhaps the SpaP_GGG, SpaP_AAA and SpaP_CCC mutants are unstable or do not assemble properly into a secretion competent EA? Similarly, the growth defect of the SpaP point mutants could also be explained by deficiencies in Sec-dependent membrane insertion.

- Supplementary Fig. 7. The EA is wrongly labelled (should be SpaPQR).

- Supplementary Fig. 19: Detailed information should be included how the pore size of the M-gate mutants has been modeled.

- It should be described in the main text what is the apo state (i.e. closed state without substrate)

- The nomenclature of the type III secretion system and injectisome should be clarified. The type III secretion system (T3SS) is the multi-component, integral membrane protein export apparatus that is responsible for secretion of effector proteins, as well as of the building blocks of the respective nanomachines (i.e. the injectisome and evolutionary related bacterial flagellum). It is confusing that the authors frequently refer to 'the T3SS' as the injectisome, although the T3SS is actually the protein export machine at the base of both the injectisome and flagellum (see also Abby and Rocha PloS Genetics 2012; Desvaux et al. Trends in Microbiology 2006).

- Line numbers would have been useful to point out specific issues in the text.

2020.12.01

Response to Referees

Manuscript: Substrate-engaged type III secretion system structures reveal mechanism for unfolded protein translocation

We would like to thank all three reviewers for carefully reading our manuscript and providing their expert opinion. We are grateful for the very constructive feedback that helped us to significantly improve our conclusions and therefore the quality and validity of the entire manuscript. We have addressed all of the referees' points below and have updated the text and figures in the revised manuscript accordingly. All changes in the manuscript can be found in the point-to-point response below and have also been highlighted in the manuscript for your convenience.

In brief, the revised manuscript now contains additional biochemical data elaborating the physiological relevance of the Q1-belt, the Q2-belt and the SpaR loop.

In very good agreement with our structural data, we find that mutations of the conserved SpaP Gln41-X-Gln43-X-Gln45 motif impairs secretion of the bacterial effector proteins SptP and SipA. Together, the structural and new biochemical data provide evidence that the hydrophilic nature of the residues located in the Q1-belt plays a major role for substrate translocation through the virulent T3SS export apparatus (EA). Further, the biochemical characterization of the conserved Gln44-Gln45 motif helped us to refine our structural conclusions in that we can now postulate that the Q2-belt plays an ancillary role in effector protein secretion.

Finally, we now provide biochemical data that further emphasizes the relevance of the SpaR loop for secretion of the effector proteins SptP and SipA. Our data provides proof that a single point mutation (SpaR I114A) substantially compromises secretion of both effector proteins. Given that substrates must first penetrate the Q1-belt and the M-gate before reaching the mutated SpaR loop site in the EA, our new data strongly supports an active contribution of the SpaR loop in the substrate translocation process.

Once again, we would like to thank all reviewers for the critical evaluation of our data, and we are looking forward to receiving your feedback on our revised manuscript.

REVIEWER COMMENTS

Reviewer #1 (Remarks to the Author):

This manuscript is well-written by Miltic et al. and presents substrate-engaged T3SS structures revealed by cryo-EM single-particle analysis to shed light on its gating mechanism for unfolded protein translocation. Two huge complex structures, apo and substrate-loaded, have been determined at enough high resolution to identify residues which involve in substrate recognition in the export apparatus (EA). In the substrate-loaded structure, the inner conduit of T3SS was occluded by an engineered effector protein that had been trapped functionally inside EA. The authors demonstrate that these structures allow us to understand a board substrate specificity in terms of the M-gate.

With proper labeling of residues in the figures, it would be helpful for general readers to gain a much better understanding of the content.

Point-to-Point response to the reviewer's comments:

This should not be in abbreviation. Here use "Serovar" instead of "sv".

Corrected in the following sentence (p.2, line 32):

'Here, we report the first structure of an active *Salmonella enterica* **serovar** Typhimurium needle complex engaged with the late effector protein SptP in two functional states, revealing the complete 800Å-long secretion conduit and unravelling the critical role of the export apparatus (EA) subcomplex in T3 secretion.'

Which is proper "Focussed or Focused"? "Focused" is used in p20. line 22 and 24.

Corrected in the following sentence (p.6; line 119):

'**Focused** refinement without any symmetry enforcement provided us with a reconstruction yielding an average resolution of ~3.3Å and resolving the entire EA, the inner rod and parts of the filament.'

"Protein Data Bank Accession Code" should be included at first appearance in the main text.

Corrected in the following sentences (p.4; 92):

'Single particle reconstruction provided us with a non-symmetrized density map of the substrate-trapped needle complex in two active functional states, ranging from 2.4 to 4.5Å in resolution (Fig. 1, Supplementary Figs. 3-5, Supplementary Tables 1-2, **Electron Microscopy Databank Accession Code: EMD-11781, Protein Data Bank Accession Code: 7ah9 and 7ahi**), and resolving a substrate density traversing through the complete secretion path from the cytoplasmic face of the needle complex to the extracellular filament (Fig. 1a,b).'

(p.6; 117):

'To be able to investigate the structural changes underlying substrate transport, we also determined the structure of a substrate-free, closed needle complex referred to herein as apo-state (**EMD-11780, 7agx**).'

Please indicate four residues included in this loop region and show labels of these residues in the figure 3 a and 3 c.

In our eyes, labeling of all residues obscures the view onto the structural features. Because we greatly appreciate the comment of the reviewer and see the benefit of better labeling for improving the clarity of the manuscript, we added a Supplementary Figure (Supplementary Fig. 12), highlighting the residues forming the Q-belts and the M-gate with labels.

Please show a label of the corresponding residue in the figure 3 a and 3c. Clearly indicate each residues with labels.

The new labels can also be found in Supplementary Fig. 12.

Does “bacterial” mean flagellar and virulent T3SSs? If this is the case, are these Gln residues conserved in FliQ of flagellar T3SSs?

To add clarity to the manuscript, we are no longer using the term bacterial T3SS. Instead, we distinguish between virulent and flagellar T3SSs.

The Gln-X-Gln-X-Gln motif in the Q1-belt of virulent T3SS is not strictly conserved in the flagellar homolog FliQ. To add clarity to the manuscript and provide further information on the similarities and differences between both systems, we added a sequence alignment of FliQ homologs (Supplementary Figure 15). Moreover, we added new biochemical data, which supports a model in which not the strict presence of Gln, but the hydrophilic nature of the side chains in general is crucial for effector protein secretion through the Q1-belt of the virulent *Salmonella* T3SS. Interestingly, hydrophilic FliQ residues Gln43, Asn45 and Glu46 are well conserved across FliQ homologs, which suggests that these residues may serve a Q1-belt like function in the structurally related flagellar system.

Accordingly, we added the following passage to the manuscript (p.7; line 141):

‘Likewise, conserved hydrophilic residues can be found in the corresponding loop connecting α -helices α 1 and α 2 in homologous flagellar FliQ, together suggesting that the Q1-belt plays an essential role for substrate translocation through virulent and flagellar T3SSs (Supplementary Fig. 15).’

Please include residue IDs of these conserved methionines in the main text.

We added residue IDs. The sentence now reads (p.8; line 167):

‘The loops between the fifth and sixth alpha helix of each of the five SpaP proteins contain three conserved **Met185-187** residues, which, together with conserved SpaR Phe212, form a hydrophobic constriction seen in substrate-free structures.’

Please include labels for the corresponding Met residues in the Fig. 4.

Please find the new labels in Supplementary Fig. 12.

Is “T3SS secretion” used properly?

Corrected. The sentence now reads (p.15; line 332):

Mutations that increase M-gate size impede cellular growth and injectisome secretion, which we believe is a result of creating leaky pores in the inner membrane, that likely hinder the proton-motive-force (PMF) believed to **power T3SSs**.

Please include residue IDs for “phenylalanines”.

We incorporated residue labels throughout the manuscript. We would like to bring to the reviewer’s attention that, based on the comment provided by Reviewer 3, the passage in question was changed to (p.15; line 339):

‘The unprecedented resolution achieved here enabled us to decipher a conformational switch of the SpaR lid which opens up to then support effector protein secretion towards the filament. In the apo-state, the conformation of the SpaR loop can be traced only with sufficiently smoothed maps and at high map thresholds in both: isolated needle complexes (our structure & 6pep) and

in the recombinant EA from *S. flexneri* (6r6b), suggesting that, in contrast to its structural counterpart in the flagellar T3SS (FliP residues: 105-124), the SpaR loop is only weakly associated with the M-gate located below (Supplementary Fig. 19). Together, our structural and biochemical data suggests that the three interfaces cooperate to facilitate substrate transport, providing a rationale as to why mutation of a single interface is not sufficient to completely prohibit effector protein secretion. In the light of this concept, it appears even more remarkable that secretion of substrates that have passed the Q1-belt and M-gate can still be compromised by a single I114A point mutation in the SpaR loop'

The missing residue IDs for FliP phenylalanines 110 and 113 can be found in Supplementary Figure 19 and have been implemented in the manuscript text as follows (p.10; line 214):

'Consistent with the recombinant export apparatus from *Shigella flexneri* injectisomes, SpaR residues Leu110 and Ile114 (Phe110 and Phe113 in FliP) interface with the methionines of the M-gate below, creating what has been termed a 'plug' in the structurally-related flagellar system (Supplementary Fig. 19 and 23)'

Please correct "sicP" to "SicP".

We corrected "sicP" to "SicP" in the manuscript.

Reviewer #2 (Remarks to the Author):

I am writing with great surprise to be reviewing the manuscript of Miletic et al ("Substrate-engaged type III secretion system structures reveal gating mechanism for unfolded protein translocation") for Nature Communications. This tour de force of structural biology, biochemistry, membrane biophysics establishes for the first time at the molecular level, with an unprecedented level of detail and mechanistic explanation, the model of non-globular polypeptide translocation through the protein Type Three Secretion Systems, with a fantastic imaging of a substrate in the process. It is the capstone on three decades of research in this area of bacterial pathogenesis, presents some astonishing EM work, and will be a powerful contribution to protein translocation in general. The work is very solid and impressively illustrated and explained. I am amazed that this is not in Nature itself as an article, especially considering the frankly underwhelming but "hot topic" manuscripts that so often pass muster for the top tiered journals. While there are minor points to quibble with, I'll let the other reviewers present such critiques. I will simply leave with the assertion that this should easily be published in Nature Communications. I will go further in fact and strongly recommend to the head editors that they go lobby to the Nature editors that this excellent and groundbreaking work be given the visibility it deserves in that journal. It would be a travesty not to do so.

We would like to thank Reviewer 2 for the deeply satisfying feedback on our structural data. The provided feedback is a fantastic acknowledgement of the years of intense work from many scientists that went into generating these structures. We agree that our work reveals - for the first time - the molecular details of the substrate translocation process through the T3SS export apparatus - data that provides a rationale to understand how non-globular protein transport across two membranes is achieved.

Reviewer #3 (Remarks to the Author):

In the present manuscript, Miletic et al. report the first high-resolution structure of a type-III secretion system (T3SS) in an active, substrate-engaged conformation. The authors further validate that a conserved methionine-rich loop at the base of the export apparatus forms a gasket during substrate protein transport in order to maintain the membrane barrier function.

The manuscript is well written and the structure of a substrate-engaged T3SS constitutes a major advance for our understanding of protein secretion via T3SS. Accordingly, the present report is well-suited for the broad readership of Nature Communications.

The experiments concerning the proposed gating mechanism largely support the author's conclusions, but some concerns remain. I have the following comments and suggestions to improve the author's conclusions and the clarity of the manuscript.

Point-to-Point response to the reviewer's comments:

The potential roles of the Q-belts and R-lid are the weakest part of the manuscript and remain largely speculative. Further functional evidence would be needed to go beyond stating mere hypotheses. In particular, concerning the SpaQ Q1-belt, only the first Q (Q41) is conserved in the flagellar T3SS (Q43), which raises questions concerning the significance of the proposed Q-belt. Similarly, the Q44Q45 residues in the SpaP Q2-belt are not conserved in FliP. It should be shown if e.g. Q-belt mutations to alanine prevent substrate translocation to support the claim that the Q-belt has a physiological role in engaging the substrate in the channel.

We would like to thank Reviewer 3 for this great idea and for pointing out a weak spot in our manuscript - the lack of biochemical data to substantiate the role of the Q-belts and SpaR loop for substrate translocation. In the revised manuscript, we now provide additional biochemical data to support the function of the Q1-belt (Figure 3d) and the Q2-belt (Fig. 5e).

With respect to the Q1-belt, these experiments demonstrate that the simultaneous mutation of all three glutamines Gln41, Gln43 and Gln45 to small aliphatic amino acids such as alanine or glycine impairs secretion of the effector proteins SptP and SipA and therefore the function of the Q1-belt. Interestingly, hydrophilic serine does not impair secretion, which strengthens the concept that a hydrophilic environment is required to facilitate substrate transport through the Q1-belt of the EA.

Accordingly, we added the following paragraph to the manuscript (p.7; line 158):

'In agreement with this concept, mutation of the three glutamines to the aliphatic amino acids alanine or glycine, but not serine, impairs secretion of SptP and SipA effector proteins, revealing that the Q1-belt plays a major role in substrate translocation (Fig. 3d).'

Further, we provide an additional sequence alignment (Supplementary Fig. 15) to highlight similarities and differences between SpaQ and its homolog in the structurally related flagellar protein FliQ. The alignment shows that, although the SpaQ Gln-X-Gln-X-Gln motif is not strictly conserved in FliQ, the loop connecting alpha helices $\alpha 1$ and $\alpha 2$ in FliQ contains seven hydrophilic amino acids (Gln39, Thr42, Gln43, Asn45, Glu46, Thr48 and Ser50), which we believe are likely to serve a similar function.

We added the following paragraph to the manuscript (p.7; line 141)

'Likewise, conserved hydrophilic residues can be found in the corresponding loop connecting α -helices α 1 and α 2 in homologous flagellar FliQ, together suggesting that the Q1-belt plays an essential role for substrate translocation through T3SSs (Supplementary Fig. 15).'

With respect to the Q2-belt, novel secretion data, that we added to the revised manuscript, reveals that mutation of the conserved Gln44-Gln45 motif to Ala44-Ala45 or Gly44-Gly45 has a moderate effect on the secretion of the effector proteins SptP and SipA (Fig. 5e). To take the refined view on the Q2-belt into account, we added the following section to the manuscript

(p.11; line 241):

'Mutation of the two SpaP glutamines Gln44 and Gln45 had only a moderate impact on SptP and SipA effector protein secretion, revealing that the Q2-belt plays an ancillary role in substrate transport, which is also in agreement with a lower conservation of these residues in the homologous flagellar protein FliP (Fig. 5e, Supplementary Fig. 17).'

Concerning the potential significance of the R-lid. The authors claim that I114 could act as the pawl of a ratchet (i.e., preventing substrates to go backwards), which should be substantiated by experimental validation. For instance, if the SpaR I114 functions a pawl that prevents substrate backward translocation, does a I114A/G mutation decrease immunoprecipitation efficiency?

Again, we would like to thank the reviewer for this valuable comment. Unfortunately, immunoprecipitation efficiency is very low already in the SB905 *Salmonella* wild-type strain, which represents the very challenge we had to overcome in this study. Therefore, in our structural experiments, we made use of a 3-times repeated SptP effector domain construct, tagged with unfoldable GFP. The main advantage of this construct over other GFP-tagged effector protein constructs is that – in the vast majority of needle complexes - the N-terminus of this artificially-lengthened construct reaches the end of the needle filament where the first SptP effector domain starts folding¹. Hence, the effector protein is effectively trapped on either side of the needle complex, a strategy which enabled us to solve the structure of a – caught in the act - needle complex, but which no longer allows to study a potential backward translocation.

However, to substantiate the role of SpaR I114 for substrate translocation, we performed effector protein secretion assays. These assays provide evidence that mutation of Ile114 to alanine is sufficient to substantially compromise SptP and SipA effector protein secretion (Fig. 5d). Given that substrates must first penetrate the Q1-belt and the M-gate before reaching the mutated SpaR loop site in the EA, our new data strongly supports an active contribution of the SpaR loop in the substrate translocation process, presumably by enhancing translocation efficiency.

Accordingly, we added/edited the following passage to/in the manuscript:

p10; line 224:

'Mutational analysis revealed that substitution of Ile114 to alanine impairs SptP and SipA effector protein secretion, indicating that the SpaR loop actively contributes to substrate transport (Fig. 5d).'

Irrespective of these new insights, we agree that our experimental data does not support a ratchet-like translocation mechanism. To no longer claim that SpaR I114 could potentially act as a pawl of a ratchet, we edited/deleted the corresponding passages in our manuscript:

(p.2; line 40):

'Following gate penetration, a moveable SpaR loop first folds up to then support substrate transport through the needle complex.'

(p.11; line 245):

~~'Together, it appears conceivable that the mobile SpaR loop functions akin to a linear ratchet in which Ile114 represents the 'pawl', whose conformational changes are physically triggered by the side chains or 'teeth' of the translocating substrate to support substrate translocation through the atrium.'~~

(p.17; line 392)

'The SpaR loop, resting above the M-gate and blocking the channel, extends upwards, assuming an open conformation, which together with the surrounding Q2-belt creates an interface to engage the substrate. 4) The upfolded SpaR loop contacts the passing substrate, and together with the Q2-belt guides the substrate up through the atrium and into the lumen of the filament.'

(p.44; line 813)

'The substrate then proceeds up to the Q2-belt (green circles) and SpaR loop (orange; Ile114 represented as black tick) which together stabilize the substrate above the M-gate and support substrate transport towards the atrium and needle filament to ultimately facilitate effector protein secretion.'

Another major concern is the effect plasmid-overproduction of SpaP point mutants (Figure 4, Supplementary Fig. 18). Does episomally expressed SpaP assemble properly in the EA complex with the correct stoichiometry?

We agree that, in principle, plasmid-based complementation of SpaP could lead to assembly defects. However, our structural data, but also the structural analyses from other homologous systems of the export apparatus (Kuhlen et al., 2018; Hu et al., 2019), shows no indication that export apparatus are structurally heterogeneous or display different stoichiometries. For all reported structures, a 5:4:1 (SpaP:SpaQ:SpaR) absolute stoichiometry has been found, indicating that proper assembly is likely under structural control and less influenced by protein levels (see also our manuscript posted on BioRxiv²).

Also, from a structural point of view, the most likely outcome of a SpaP overexpression scenario is the formation of SpaP hexamers, in which a sixth SpaP copy would replace SpaR. While we cannot rule out that such SpaP hexamers indeed form, the absence of SpaR in putative hexameric SpaP complexes would inevitably result in the termination of needle complex assembly as knockout experiments provide evidence that SpaR is indispensable for this process. Consequently, the expectation would be that plasmid-complementation of SpaP should result in fewer needle complexes and lower effector protein secretion levels.

However, we found that secretion of the two substrates SptP and SipA, with episomally expressed SpaP (wild-type sequence) in a SpaP knockout strain (Δ SpaP) shows similar (or slightly increased) levels as assessed by Western blotting, compared to chromosomally-encoded SpaP in our SB905 wild-type strain). We are convinced, this data validates the experimental set-up (Figure 5e).

Figure 5e. Representative immunoblots of secretion assays of SptP and SipA effector proteins using wild-type SB905, SpaP knockout (Δ SpaP) and Δ SpaP strains complemented with wild-type SpaP (WT CP). Adapted from Figure 5e in the manuscript.

I would assume that SpaQ and SpaR, which are still under their native physiological expression control might not be present in the corresponding levels?

To address the raised concerns for our plasmid-borne reconstitution experiments, mutations in SpaQ and SpaR have been introduced into the bacterial chromosome. Hence, we do not expect different expression levels of these proteins.

Accordingly, any growth differences would be much more convincing if the SpaP mutants would be expressed under native, physiological conditions.

We kindly refer to our episomal SpaP expression-related comment above.

Further, significant growth differences between the vector control, the WT SpaP and the SpaP point mutants are unclear as also the vector control shows different growth in the presence or absence of inducer?

We agree that the observed growth differences may appear small. However, we would like to bring to the reviewers attention that our growth experiments reveal a general effect upon addition of the needle complex inducer arabinose - which is - a moderate increase in OD_{600} values over the time course of the experiment only in those strains that do not suffer from leaky pore formation (reconstituted wild-type SpaP (WT CP), SpaP^{WWW} and SpaP^{FFF}). In contrast, strains in which the M-gate has been manipulated to form leaky membrane pores (SpaP^{CCC}, SpaP^{AAA}, SpaP^{GGG}), show a small but robust impairment in growth as judged from the analysis of 9 independently performed experiments.

Additional comments:

- Page 8-9: It has already been shown previously for FliP, the SpaP homolog of the flagellar T3SS that the M-gate_AAA mutation, but not the M-gate_FFF mutation induces a strong membrane permeability (Ward et al. Mol Microbiol, 107: 94–103). This should be discussed.

We implemented the reviewers' suggestion as follows (p.9; line 196):

'Consistent with this hypothesis, replacement of the corresponding methionines (Met209-211) in the flagellar homolog FliP to alanines has been shown to increase membrane conductance.'

Accordingly, we also edited the following paragraph in the discussion (p.15; line 330):

'Our data confirms the M-gate forms the main constriction in apo-state structures that shifts just enough to allow substrate passage while likely preventing unwanted transport of other molecules. Mutations that increase M-gate size impede cellular growth and injectisome secretion, which we believe is a result of creating leaky pores in the inner membrane, that likely hinder the proton-motive-force (PMF) believed to power T3SSs. Consistent with this hypothesis, deletion of one of the M-gate methionines or mutation to less bulky residues in the flagellar homolog FliP increases ion conductance, while mutation to bulkier phenylalanine residues does not. Taken together, this suggests that the M-gate methionines are critical residues providing flexibility for substrate transport while forming a selective barrier.'

- Page 15: What is the evidence that the flagellar EA rotates, hence necessitating the conserved phenylalanine residues in FliR? Even if the flagellar EA rotates, it is unclear if centrifugal forces play a role at these size scales?

We thank the reviewer for pointing this out. We removed the questionable passage from the discussion of the manuscript to no longer claim that the flagellar EA rotates.

- Page 15: Evidence from the flagellar T3SS suggests that substrate proteins are likely secreted in an alpha-helical conformation (e.g. as shown for the hook length control protein FliK; Shibata et al. Mol Microbiol, 64: 1404–1415).

We have implemented the requested reference into our discussion, which now reads (p.17; line 370):

'Also, helical filament assembly could be more effectively achieved by docking of the readily formed helices into their cognate interfaces at the distal end of the growing filament. This concept is in agreement with measurements revealing the hook length control protein FliK is being transported through flagellar T3SS in alpha helical form'.

- Figure 4: Why are the leaky M-gate mutants impaired in substrate protein transport? Perhaps the SpaP_GGG, SpaP_AAA and SpaP_CCC mutants are unstable or do not assemble properly into a secretion competent EA? Similarly, the growth defect of the SpaP point mutants could also be explained by deficiencies in Sec-dependent membrane insertion.

Why are the leaky M-gate mutants impaired in substrate protein transport?

This is a very intriguing question. In our opinion, Supplementary Figure 20 gives a partial explanation for the question raised. Purification of needle complexes from the SpaP^{AAA} mutant *Salmonella* SB905 strain is possible and negative stain electron microscopy confirms the presence of fully-assembled needle complexes, albeit at lower numbers compared to wild-type SB905. The presence of the needle filament itself provides evidence that both: the SpaP^{AAA} and

SpaP^{CCC} mutant strains retain at least some ability to secrete the rod and needle filament proteins PrgI and PrgJ through their mutated export apparatus. In contrast, fully-assembled needle complexes are absent in strains lacking SpaR, SpaP or SpaQ expression³ as a direct consequence of the failure to assemble a functional, secretion-competent export apparatus.

The current model of substrate translocation requires the concerted action of the hexameric T3SS ATPase InvC and the proton motive force (PMF). However, deciphering the exact contribution of the PMF to T3SS effector protein secretion has remained a very delicate task, first and foremost because manipulating the PMF has detrimental consequences affecting all ATP-dependent cellular processes. Importantly, mutation of the MMM-motif in the flagellar homolog FlhP - a study that we refer to in the discussion of our manuscript - causes increased ion flux, a result that is in very good agreement with the concept of leaky pore formation through the assembly of mutated EAs.

Therefore, we believe that the herein reported growth defect - which can only be observed under injectisome-inducing conditions in SpaP^{CCC}, SpaP^{AAA} and SpaP^{GGG} mutant SB905 strains, strongly implies the formation of leaky membrane pores that impair the PMF and consequently T3SS effector protein secretion.

Perhaps the SpaP_GGG, SpaP_AAA and SpaP_CCC mutants are unstable or do not assemble properly into a secretion competent EA?

We did not investigate SpaP protein stability in our study and therefore we cannot give a direct answer to this question. However, our data reveals that mutation of the MMM-motif in SpaP to FFF restores needle complex formation and functionality upon complementation of a SpaP knockout strain and therefore replacement of the MMM-motif per se does not render SpaP more unstable (Fig. 4).

Further, all structural information available on virulent EAs demonstrate that formation of the pentameric SpaP subcomplex is driven by an extensive interface that results from the packing of SpaP alpha helices $\alpha 6$, $\alpha 7$ and $\alpha 8$ in one SpaP protomer against alpha helices $\alpha 3$, $\alpha 10$ and $\alpha 11$ of its SpaP neighbor. The MMM-motif however resides in a flexible loop connecting alpha helices $\alpha 8$ and $\alpha 9$ and therefore - from a structural point of view - a direct impact on EA complex formation is very difficult to comprehend.

Similarly, the growth defect of the SpaP point mutants could also be explained by deficiencies in Sec-dependent membrane insertion.

To the best of our knowledge, very little is known about EA protein signal peptides and the mechanisms by which EA proteins are being inserted into the inner bacterial membrane. We agree with the reviewers comment in that we cannot rule out that our mutations may compromise membrane protein insertion, however given that the mutations are not placed in a transmembrane helix directly but located in a neighboring loop, we believe membrane protein insertion deficiencies are unlikely to happen.

Also, all mutations placed (WWW, FFF, AAA, GGG) - with cysteine being the only exception - maintain the general hydrophobic nature of the methionine side chain and therefore represent minimal chemical manipulations to the SpaP primary sequence.

Finally, we are convinced that a failure in membrane protein insertion would abrogate EA assembly in the inner bacterial membrane. Consequently, no leaky membrane pores would form in the inner bacterial membrane and an impaired bacterial growth would not be detectable - as

can be seen for SpaP knockout strains complemented with SpaP^{WWW} and SpaP^{FFF} mutant proteins.

In agreement with this concept, replacement of the FliP MMM-motif with alanines (FliP^{AAA}), but not phenylalanines (FliP^{FFF}), has been reported to increase guanidinium conductance in the structurally related flagellar system⁴.

- Supplementary Fig. 7. The EA is wrongly labelled (should be SpaPQR).

Thank you for pointing this out. The EA labels have been corrected.

- Supplementary Fig. 19: Detailed information should be included how the pore size of the M-gate mutants has been modeled.

Hole sizes were measured with ChimeraX. A section explaining the procedure has been added to the Material and Methods section (p.24; line 533):

'Putative pore sizes of SpaPCCC-, SpaPAAA- and SpaPGGG-containing EAs were determined using the mark surface in combination with the distance measurement tool in ChimeraX.'

- It should be described in the main text what is the apo state (i.e. closed state without substrate)

We now introduce the substrate-free apo-state as follows (p.6; line 117):

'To be able to investigate the structural changes underlying substrate transport, we also determined the structure of a substrate-free, closed needle complex referred to herein as apo-state (EMD-11780, 7agx).'

- The nomenclature of the type III secretion system and injectisome should be clarified. The type III secretion system (T3SS) is the multi-component, integral membrane protein export apparatus that is responsible for secretion of effector proteins, as well as of the building blocks of the respective nanomachines (i.e. the injectisome and evolutionary related bacterial flagellum). It is confusing that the authors frequently refer to 'the T3SS' as the injectisome, although the T3SS is actually the protein export machine at the base of both the injectisome and flagellum (see also Abby and Rocha PloS Genetics 2012; Desvaux et al. Trends in Microbiology 2006).

We apologize for the confusion caused by using T3SS and injectisome as synonyms. We adjusted the use according to the reviewers' suggestion. To further improve clarity, we added Supplementary Table 2 listing the unifying T3SS nomenclature.

- Line numbers would have been useful to point out specific issues in the text.

We apologize for the inconvenience. Line numbers have been added to the revised manuscript.

References

1. Radics, J., Königsmäier, L. & Marlovits, T. C. Structure of a pathogenic type 3 secretion system in action. *Nat. Struct. Mol. Biol.* **21**, 82–87 (2014).

2. Goessweiner-Mohr, N. *et al.* *Structural control for the coordinated assembly into functional pathogenic type-3 secretion systems*. <http://biorxiv.org/lookup/doi/10.1101/714097> (2019) doi:10.1101/714097.
3. Sukhan, A., Kubori, T., Wilson, J. & Galán, J. E. Genetic analysis of assembly of the *Salmonella enterica* serovar Typhimurium type III secretion-associated needle complex. *J. Bacteriol.* **183**, 1159–67 (2001).
4. Ward, E. *et al.* Type-III secretion pore formed by flagellar protein FliP. *Mol. Microbiol.* **107**, 94–103 (2018).

REVIEWERS' COMMENTS

Reviewer #1 (Remarks to the Author):

This revised manuscript of "Substrate-engaged type III secretion system structures reveal gating mechanism for unfolded protein translocation" by Miletic et al. is improved significantly and feasible to understand the molecular mechanism of the type III secretion apparatus underpinned by the high-resolution CryoEM structures. Principally, I could not find any comments or further claims. I think this manuscript will be accepted without any additional corrections and strongly suggest to publish it in Nature Communications.

Reviewer #3 (Remarks to the Author):

Miletic et al. extensively revised the manuscript and I would like to congratulate the authors for this exciting work.

The additional biochemical experiments substantially improve the manuscript and nicely support the authors' conclusions for the most parts. I have a few remaining remarks that the authors might consider discussing in order to clarify some open questions.

- Fig. 3d: It is unclear why secretion levels of the Q1-belt A-A-A mutant are closer to WT than the G-G-G mutant. This might suggest an (additional) defect in assembly of the G-G-G mutant, which could be discussed as possibility in the text.

Of note, FliQ_M47 (homolog of SpaQ_Q45) appears to be rather tolerant to substitutions (doi: 10.1101/2020.11.25.397760).

Line 143 should therefore rather read: "suggesting that the Q1-belt could play an important role for substrate translocation" and line 160 "revealing that the Q1-belt is implicated in substrate translocation".

- Fig. 5d/e: Both the Q2-belt AA/GG and the SpaR I114A mutants appear to only show a moderate effect on secretion. This rather indicates an ancillary role of both the SpaR loop and Q2-belt for substrate transport. Quantification of the Western blot results would perhaps be needed to substantiate any minor differences.

- In respect to the potential role of the Q1-belt and R-lid in substrate translocation efficiency, it might be relevant to discuss a possible, more global role in pore closing and pore gating, e.g. as suggested previously by Kuhlen et al. (doi: 10.1038/s41594-018-0086-9) who state: "The Q-latch is more open and hydrophilic than either the P-gasket or the R-plug, but it wraps around the P subunits and holds them closed via highly conserved interchain salt bridges. The tight closure of the PQR complex is likely to be important for maintaining bacterial viability, ensuring the complex does not form breaches in the cell membrane before full assembly of the basal body and control of gating by the entire T3SS."

2020.12.28

Response to Referees (Revision)

Manuscript: Substrate-engaged type III secretion system structures reveal mechanism for unfolded protein translocation

We would like to thank both reviewers for providing feedback and additional comments to our revised manuscript. In the new revision we have incorporated all points. A detailed point-to-point response is provided below and all changes have been highlighted in the revised manuscript for your convenience.

Comments to revised manuscript:

Reviewer #1 (Remarks to the Author):

This revised manuscript of “Substrate-engaged type III secretion system structures reveal gating mechanism for unfolded protein translocation” by Miletic et al. is improved significantly and feasible to understand the molecular mechanism of the type III secretion apparatus underpinned by the high-resolution CryoEM structures. Principally, I could not find any comments or further claims. I think this manuscript will be accepted without any additional corrections and strongly suggest to publish it in Nature Communications.

We would like to thank reviewer #1 again for the comments on the first submitted version, which allowed us to clarify some points and improve the content of figures. We see the feedback on the revised version as well as the recommendation to publish this work in Nature Communication not only as a great opportunity to share our scientific findings with the scientific community but also as a great appreciation for those colleagues who have carried out this work with great care and dedication over many years.

Reviewer #3 (Remarks to the Author):

Miletic et al. extensively revised the manuscript and I would like to congratulate the authors for this exciting work.

The additional biochemical experiments substantially improve the manuscript and nicely support the authors' conclusions for the most parts.

We would like to thank reviewer #3 again for the comments on the first submitted version, which allowed us to revise our manuscript by adding additional functional data. Together with the structural data, we demonstrate the importance of the three-way interface within the export apparatus for substrate transport in T3SSs.

I have a few remaining remarks that the authors might consider discussing in order to clarify some open questions.

- Fig. 3d: It is unclear why secretion levels of the Q1-belt A-A-A mutant are closer to WT than the G-G-G mutant. This might suggest an (additional) defect in assembly of the G-G-G mutant, which could be discussed as possibility in the text.

Of note, FliQ_M47 (homolog of SpaQ_Q45) appears to be rather tolerant to substitutions (doi:

10.1101/2020.11.25.397760).

Line 143 should therefore rather read: "suggesting that the Q1-belt could play an important role for substrate translocation" and line 160 "revealing that the Q1-belt is implicated in substrate translocation".

We now discuss the possibility of assembly defects (line 179) and provide information to FliQ (line 162). Moreover, we have added an additional supplementary figure showing a multiple sequence alignment highlighting the conserved hydrophilic "Q1-like-belt" with residues in FliQ (including M47) and homologs.

Line 162

Likewise, conserved hydrophilic residues can be found in the corresponding loop connecting α -helices $\alpha 1$ and $\alpha 2$ in homologous flagellar FliQ, together suggesting that the Q1-belt could play an important role for substrate translocation through virulent and flagellar T3SSs

Line 179

In agreement with this concept, mutation of the three glutamines to the aliphatic amino acids alanine or glycine, but not serine, impairs secretion of SptP and SipA effector proteins, revealing that the Q1-belt is implicated in substrate translocation (Fig. 3d). In addition, mutation of the Q1-belt could cause assembly defects as seen in the different secretion profiles of the alanine and glycine mutants

- Fig. 5d/e: Both the Q2-belt AA/GG and the SpaR I114A mutants appear to only show a moderate effect on secretion. This rather indicates an ancillary role of both the SpaR loop and Q2-belt for substrate transport. Quantification of the Western blot results would perhaps be needed to substantiate any minor differences.

We agree that the mutations in Q2 show a moderate yet stable effect on secretion, whereas the mutations in SpaR (I114A) has a stronger impact especially to SipA. We have thus added the following sentences as well as have quantified SptP and SipA secretion (Fig 5).

Line 255

Mutational analysis revealed that substitution of Ile114 to alanine impairs SptP and SipA effector protein secretion, indicating that the SpaR loop actively contributes to substrate transport (Fig. 5d,e).

Line 272

Mutation of the two SpaP glutamines Gln44 and Gln45 had only a moderate impact on SptP and SipA effector protein secretion, revealing that the Q2-belt plays an ancillary role in substrate transport, which is also in agreement with a lower conservation of these residues in the homologous flagellar protein FliP

- In respect to the potential role of the Q1-belt and R-lid in substrate translocation efficiency, it might be relevant to discuss a possible, more global role in pore closing and pore gating, e.g. as suggested previously by Kuhlen et al. (doi: 10.1038/s41594-018-0086-9) who state: "The Q-latch is more open and hydrophilic than either the P-gasket or the R-plug, but it wraps around the P subunits and holds them closed via highly conserved interchain salt bridges. The tight closure of the PQR complex is likely to be important for maintaining bacterial viability, ensuring the complex does not form breaches in the cell membrane before full assembly of the basal body and control of gating by the entire T3SS."

We added the following sentence in the discussion and provide the above mentioned citation.

Line380

Together, our structural and biochemical data suggests that the three interfaces cooperate to facilitate substrate transport, providing a rationale as to why mutation of a single interface is not sufficient to completely prohibit effector protein secretion. Conceivably the SpaR lid, the SpaP M-gate and the SpaQ Q1-belt, function together to gate the needle complex channel. The SpaQs securely support the above SpaP subunits and SpaR, creating the tight gasket in the channel needed for maintaining membrane homeostasis as suggested in the flagellar EA¹². In the light of these concepts, it appears even more remarkable that secretion of substrates that have passed the Q1-belt and M-gate can still be compromised by a single I114A point mutation in the SpaR loop

Once again, we would like to thank all three reviewers for the critical evaluation of our data.